# STABLE: Simulation-Ready Tabletop Layout Generation via a Semantics–Physics Dual System

Zhen Luo [* 1 2]   Yixuan Yang [* 1 3]   Xudong Xu [† 3]   Jinkun Hao [4]   Zhaoyang Lyu [3]   Feng Zheng [† 1 5]
Jiangmiao Pang [3]   Yanwei Fu [2 6]

## Abstract

Generating simulation-ready tabletop scenes from task instructions is an intriguing and promising research direction in the field of Embodied AI. However, existing task-to-scene generation methods rely exclusively on large language models (LLMs) to predict scene layouts, inevitably yielding object collisions or floating due to LLMs' inherent limitations in 3D spatial reasoning. In this paper, we present **STABLE**, a semantics–physics dual-system tailored for simulation-ready tabletop scene generation. STABLE consists of two complementary modules: (i) a **Semantic Reasoner**, a fine-tuned LLM trained on a structured tabletop scene dataset to generate coarse layouts from input task instructions, and (ii) a **Physics Corrector**, a physics-aware flow-based denoising model that outputs pose updates to refine layouts, which ensures the physical plausibility of scenes while preserves semantic alignment with task instructions. STABLE adopts a progressive generation paradigm: by alternating between the Semantic Reasoner and Physics Corrector, it incrementally expands the scene from task-critical objects to background objects. Experiments demonstrate that STABLE successfully generates simulation-ready tabletop scenes that strictly conform to task instructions and significantly enhances the physical validity of scenes over prior art.

## 1. Introduction

Synthetic data is emerging as an increasingly vital component in both the training and evaluation phases of embodied AI, thanks to its inherent advantages of low cost and

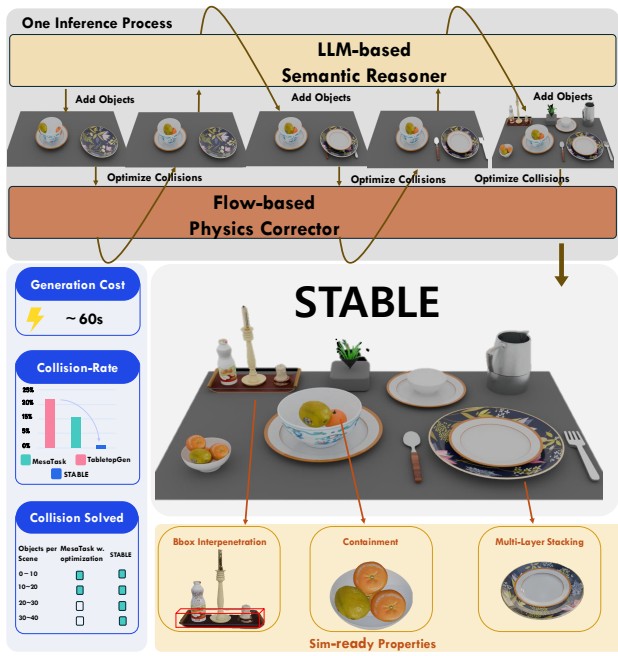

*Figure 1.* Overview of our proposed **STABLE** for tabletop scene generation. STABLE is a Semantics–Physics dual system that alternates between an LLM-based Semantic Reasoner and a geometry-aware flow-based Physics Corrector to generate diverse, task-aligned, and simulation-ready tabletop layouts.

easy scalability (Deitke et al., 2022; Tian et al., 2025). For synthetic data in robotic manipulation, the generation of diverse tabletop scenes that align with manipulation task instructions can provide a wide range of robotic simulation environments, thereby receiving increasing attention recently (Gao et al., 2025; Chen et al., 2025). Undoubtedly, the ability to effectively interpret high-level task instructions while ensuring the generation of physically plausible or simulation-ready scenes has become the key to this generation task.

In recent years, researchers have attempted to leverage powerful large language models (LLMs) to interpret text descriptions or task instructions for the purpose of scene generation. A growing body of LLM-based scene generation methods—whether directly harnessing the zero-shot capabil-

---

[*]Equal contribution   [†]Corresponding author. [1]SUSTech [2]SII [3]Shanghai AI Laboratory [4]SJTU [5]Spatialtemporal AI [6]FDU. Correspondence to: Zhen Luo <luoz2024@mail.sustech.edu.cn>.

*Proceedings of the 43rd International Conference on Machine Learning*, Seoul, South Korea. PMLR 306, 2026. Copyright 2026 by the author(s).

ities of LLMs (Feng et al., 2023; Yang et al., 2026), adopting multi-step prompting strategies (Çelen et al., 2024; Yang et al., 2024; Sun et al., 2025; Yang et al.), or further fine-tuning LLMs on scene datasets (Yang et al., 2025a;b; Hao et al., 2026)—are capable of translating high-level semantics into structured scene layouts. Despite yielding promising generation results for both room-scale and tabletop scenes, these methods still struggle to synthesize simulation-ready scenes (Çelen et al., 2024; Wang et al., 2025; Hao et al., 2026), often resulting in object interpenetration or floating artifacts. We attribute these failures to the hallucination phenomenon of LLMs as well as their inherent limitations in 3D spatial reasoning. Meanwhile, certain post-hoc optimization attempts can improve the physical plausibility of generated scenes to some extent (Pfaff et al., 2025; Yao et al., 2025). However, such optimizations inevitably impose heavy computational burdens and even fail to find a feasible solution for scenes afflicted with severe object collisions. More importantly, they may disrupt the original scene layouts, thereby resulting in cluttered or out-of-distribution layouts and potentially undermining alignment with the given task instructions (Hao et al., 2026; Pfaff et al., 2025). For instance, if the task instruction specifies placing an apple to the left of a banana on the tabletop, such post-hoc optimizations may erroneously displace the apple to the banana's right.

Inspired by the impressive "System 1, System 2" VLA model adopted in Helix (Figure, 2024), we propose **STABLE**, a Semantics–Physics dual-system framework for generating task-aligned and simulation-ready tabletop scenes without compromising scene layout diversity. Specifically, STABLE integrates two complementary components: a **Semantic Reasoner** and a **Physics Corrector**. Following prior works(Hao et al., 2026), the Semantic Reasoner is an LLM fine-tuned on MesaTask-10K via supervised fine-tuning (SFT). Given a task instruction, the Semantic Reasoner is responsible for interpreting the semantics of the instruction and generating a coarse yet task-aligned scene layout that specifies task-relevant objects and their interrelations. The Physics Corrector is a lightweight flow-based denoising model that predicts pose updates to refine the scene layout. To ensure the physical plausibility of the generated scenes, we deliberately equip the Physics Corrector with explicit geometric awareness, *i.e.*, conditioning it on object point clouds to predict object pose updates. Moreover, we employ subtle signed distance function (SDF) collision losses, which are highly sensitive to inter-object penetration, to effectively resolve object-object and object-table collisions. Additionally, a support-contact loss is utilized to eliminate object floating artifacts in the generated scenes. The Physics Corrector is also trained on MesaTask-10K(Hao et al., 2026), enabling it to learn a reasonable layout distribution of tabletop scenes and thus avoid generating cluttered

tabletop layouts—even when handling coarse layouts with severe collision issues.

For tabletop layout synthesis, STABLE leverages a progressive inference workflow characterized by alternating iterations of the Semantic Reasoner and Physics Corrector. Specifically, the Semantic Reasoner incrementally expands the layout from task-critical objects to contextual background objects, with the Physics Corrector invoked immediately after each expansion stage to preserve the physical plausibility of the scene. This design not only mitigates error accumulation that would otherwise lead to severe object collisions but also inherently enables incremental scene editing and arrangement.

We summarize our main contributions as follows: (1) We propose STABLE, a first-of-its-kind dual-system framework for scene generation, which directly maps task instructions to simulation-ready tabletop layouts by decoupling semantic layout generation from physics-aware pose correction. (2) We introduce a geometry-aware denoising target, which allows the Physics Corrector to handle complex or challenging spatial relationships such as stacking and containment. (3) Extensive experiments demonstrate that STABLE can generate simulation-ready tabletop scenes with strict alignment to task instructions, achieving substantial improvements in both the physical validity and task alignment of scenes compared to existing methods.

## 2. Related Work

### 2.1. 3D Layout Generation

Recent advances in 3D scene layout generation can be broadly grouped into (i) prompting-driven approaches built on proprietary foundation models and (ii) data-driven approaches trained on curated 3D layout datasets. Prompting-driven methods(Feng et al., 2023; Littlefair et al., 2025; Çelen et al., 2024; Yang et al., 2024; Sun et al., 2025; Yang et al.; 2026) typically rely on large closed-source LLMs and carefully designed prompting rules or multi-round interactions to elicit spatial reasoning and produce structured layouts. While these methods demonstrate strong semantic priors, they largely operate at the textual level and provide limited mechanisms for obtaining geometry-aware feedback from the generated layouts, making it difficult to consistently enforce physical plausibility. A related line of work(Wang et al., 2025; 2024; Huang et al., 2025), introduces intermediate image representations to bridge text and 3D scenes, using vision signals to help models interpret coordinates and spatial arrangements. However, such long-horizon pipelines often suffer from error accumulation and increased cost, and can be especially brittle under cluttered tabletop scenes with occlusions and stacking. On the other hand, the availability of large-scale 3D datasets

such as 3D-FRONT(Fu et al., 2021) and MesaTask(Hao et al., 2026) has enabled data-driven learning of layout distributions. Methods like LlPlace(Yang et al., 2025a), OptiScene(Yang et al., 2025b), and MesaTask(Hao et al., 2026) improve open LLMs via supervised fine-tuning to generate 3D layouts from language. Nevertheless, LLMs still struggle with continuous pose generation: discretized numbers can lead to inaccurate coordinates, resulting in collisions, interpenetration, or floating objects. Meanwhile, denoising-based layout generators(Tang et al., 2024; Wei et al., 2023; Lin & Yadong; Liu et al., 2022) can better model continuous variables, but often exhibit limited generalization when tested on diverse layouts and unseen object configurations. These observations suggest the need for models that better couple semantic layout generation with explicit geometric and physical constraints.

## 2.2. Physical Optimization

By optimizing the scene to make unreasonable phenomena conform to physical laws, we can avoid problems such as objects flying away or other scene-breaking issues that often occur when directly using a physics engine to enforce constraints on the generated scene. Recent methods incorporate such principles either by running post-hoc physical optimization or simulation for generated layouts, or by embedding physics-aware objectives into the generation pipeline. Some methods(Yao et al., 2025; Pfaff et al., 2025) applies physics-based post-processing to guarantee feasible cluttered scenes, while PAT3D(Lin et al., 2025) introduces physics-in-the-loop augmentation for text-to-3D generation to improve stability and simulation usability. These approaches are effective in improving physical feasibility, but often introduce additional computation due to iterative optimization or simulator-in-the-loop procedures, especially for cluttered scenes with complex contacts. In this work, we explore an alternative direction that enforces physical feasibility through a learned pose correction module with mesh-level constraints, which complements existing physics-based optimization and simulation pipelines.

## 3. Method

### 3.1. Problem Formulation

We study task-oriented tabletop layout generation, where the goal is to produce a simulation-ready tabletop scene conditioned on a natural language instruction. Given a task instruction $I$ and a tabletop specification $T$, we aim to generate a scene $S$ that can be directly used for physical simulation.

We represent $S$ with a structured layout description (stored as a JSON file)

$$J = \left\{ T, \{O_i\}_{i=1}^N \right\}, \tag{1}$$

where $T$ specifies the tabletop (e.g., its size), and each object is parameterized by

$$O_i = \{\mathbf{p}_i, \ r_i, \ s_i, \ d_i\}. \tag{2}$$

Here $\mathbf{p}_i \in \mathbb{R}^3$ denotes 3D translation, $r_i \in \mathbb{R}$ is the yaw rotation around the vertical axis, $s_i \in \mathbb{R}^3$ is the bounding-box size, and $d_i$ is a textual description of category/shape/appearance. Each object is additionally associated with an asset identifier $a_i$ retrieved from a 3D asset library using $(s_i, d_i)$, which provides mesh geometry for physics-aware modeling.

Our framework decomposes generation into two complementary modules: a Semantic Reasoner (SR) that predicts a coarse, task-grounded layout, and a Physics Corrector (PC) that enforces physical feasibility by updating only object poses. Formally, SR produces an initial layout $J$ and PC outputs pose updates:

$$J \leftarrow \mathrm{SR}(I, T), \qquad \{(\mathbf{p}_i, r_i)\}_{i=1}^N \leftarrow \mathrm{PC}(J). \tag{3}$$

The final scene $S$ is then assembled from $J$ using the retrieved assets and the updated poses.

### 3.2. Semantic Reasoner: Progressive Task-Grounded Layout Generation

Different from the original MesaTask(Hao et al., 2026) pre-processing pipeline, we tailor the instruction-to-layout representation to better support progressive scene construction in our dual-system framework. To keep inference efficient, we remove long-form reasoning chains and train the LLM to directly output structured layouts, substantially reducing token length and generation latency.

To enable progressive generation at inference time, we convert each MesaTask-10K layout into a serialized sequence of sub-layouts aligned to the same task instruction. Given a scene with object set $\mathcal{O}$, we partition it into three groups: task-oriented objects $O^t$, important background objects $O^B$, and secondary background objects $O^b$.

The Semantic Reasoner constructs the full layout in three stages:

$$O^t \leftarrow \mathrm{SR}(I, T), \tag{4}$$
$$O^B \leftarrow \mathrm{SR}(I, T, O^t), \tag{5}$$
$$O^b \leftarrow \mathrm{SR}(I, T, O^t, O^B), \tag{6}$$

and the complete object set is $\mathcal{O} = O^t \cup O^B \cup O^b$.

Task-oriented objects $O^t$ come directly from MesaTask-10K annotations and correspond to entities explicitly specified in the instruction, which are indispensable for task execution. We define important background objects $O^B$ as objects strongly coupled with $O^t$ in the final scene, i.e., objects that

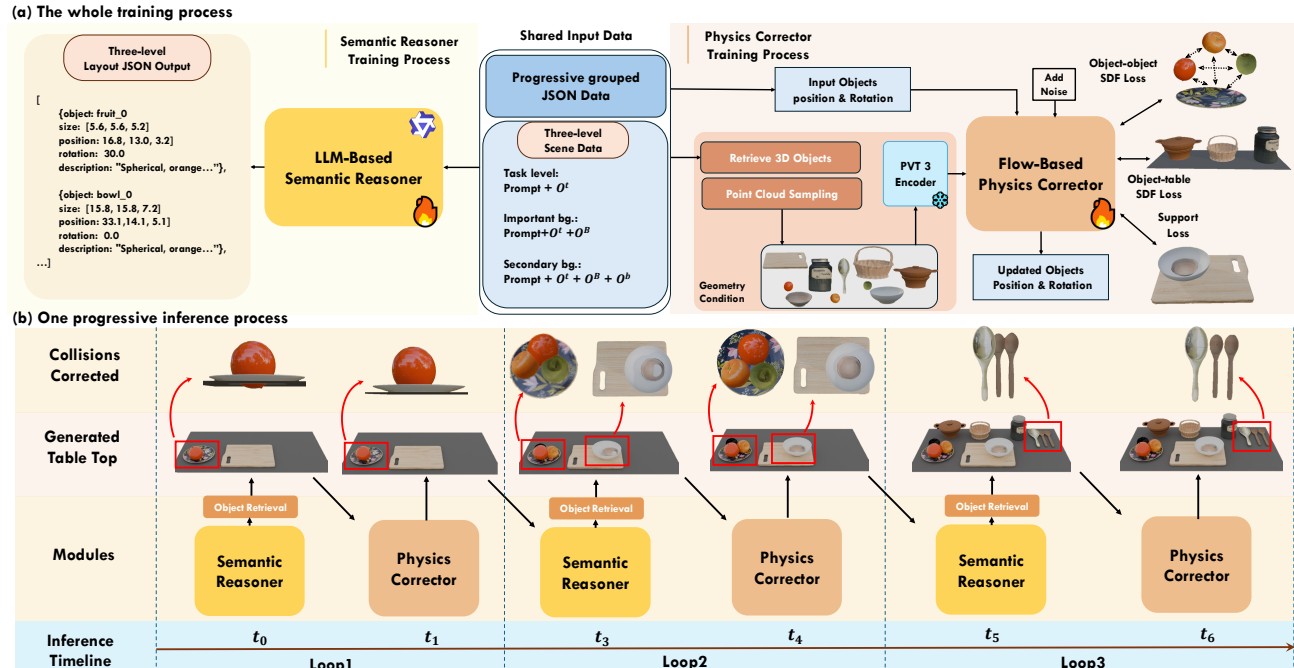

*Figure 2.* **Overview of our proposed framework. (a) The training pipeline.** Our framework decomposes task-oriented layout generation into two decoupled modules. The **Semantic Reasoner** (left) is an LLM-based model trained to generate structured JSON layouts progressively in three levels: task-oriented objects ($O^t$), important background ($O^B$), and secondary background ($O^b$). The **Physics Corrector** (right) is a flow-based generative model designed to refine object poses (translation and rotation). It is conditioned on geometry embeddings extracted from surface point clouds of retrieved 3D assets (via a frozen PVT-3D encoder). To ensure simulation readiness, the corrector is trained with mesh-level SDF collision losses (object-object and object-table) and a support stability loss. **(b) The progressive inference process.** STABLE alternates a Semantic Reasoner and a Physics Corrector across stages (Loop 1–3): the Reasoner expands the layout semantics, assets are retrieved accordingly, and the Corrector refines object poses to remove physical violations (red boxes) before feeding a simulation-ready sub-layout to the next stage.

are in physical contact with or placed very close to any task-oriented object. In practice, we identify $O^B$ by selecting objects whose 3D bounding boxes intersect with that of at least one object in $O^t$ (within a predefined threshold). The remaining objects are labeled as secondary background objects $O^b$.

This serialization provides two benefits: (i) it strengthens instruction grounding by explicitly separating instruction-critical objects from background clutter, reducing the risk of missing core objects; and (ii) it improves SR's ability to progressively complete cluttered scenes, which aligns naturally with our dual-system inference and facilitates incremental scene editing.

### 3.3. Physics Corrector: Physics-aware Flow-based Pose Correction

Given a coarse layout proposed by the Semantic Reasoner, the overall object set and spatial relations are typically semantically reasonable; however, due to the LLM's limited ability to model continuous geometry, the resulting poses often exhibit non-physical artifacts such as collisions, interpenetration, and floating objects. The Physics Corrector (PC) is designed to learn a physically consistent pose gener-

ation process in continuous space. Concretely, PC parameterizes a pose update model with a U-Net backbone and predicts updated translations and yaw rotations $\{(\mathbf{p}_i, r_i)\}_{i=1}^N$, while keeping object identities, sizes $\{s_i\}$, and semantic descriptions $\{d_i\}$ fixed. This design enforces physical validity without altering the semantic structure or distorting the scene distribution established by the Semantic Reasoner.

**Geometry-aware conditioned Generation.** Relying solely on 3D bounding-box information is often insufficient to handle complex tabletop relationships (e.g., stacking and containment), since a box is a coarse approximation of true object geometry. To provide shape cues, we condition the Physics Corrector on mesh-level geometric features of each retrieved asset. Specifically, for object $O_i$ we retrieve its asset $a_i$, scale it to match the predicted size $s_i$, sample a surface point cloud $\mathcal{P}_i$ from the scaled mesh, and extract a geometry embedding using a frozen point-cloud encoder PointTransformerV3(Wu et al., 2024) $\phi(\cdot)$:

$$\mathbf{g}_i = \phi(\mathcal{P}_i), \qquad \mathbf{G} = \{\mathbf{g}_i\}_{i=1}^N. \qquad (7)$$

We use $\mathbf{G}$ as conditioning inputs to guide pose generation, enabling the model to resolve collisions and produce physically plausible updates under complex contact patterns.

**Flow Matching for pose correction.** We perform continuous pose refinement only on each object's 3D position and 1D yaw, while keeping identities, sizes, and semantics fixed. Let the pose vector concatenate all objects' states:

$$\mathbf{x} = \begin{bmatrix} \mathbf{p}_1, \dots, \mathbf{p}_N, \ r_1, \dots, r_N \end{bmatrix} \in \mathbb{R}^{4N}, \qquad (8)$$

where $\mathbf{p}_i \in \mathbb{R}^3$ and $r_i \in \mathbb{R}$ denote the translation and yaw of object $i$. Given a coarse pose $\mathbf{x}^c$ from the Semantic Reasoner and geometry conditioning $\mathbf{G}$, we define $\mathcal{C} = (\mathbf{x}^c, \mathbf{G})$ and train a conditional velocity field $\mathbf{v}_\theta(\mathbf{x}_t, t, \mathcal{C})$ (with a U-Net backbone) using Flow Matching(**?**). Concretely, we add Gaussian noise to the coarse pose in the same $(\mathbf{p}, r)$ space:

$$\mathbf{x}_0 = \mathbf{x}^c + \sigma\boldsymbol{\epsilon}, \quad \boldsymbol{\epsilon} \sim \mathcal{N}(\mathbf{0}, \mathbf{I}), \qquad \mathbf{x}_1 = \mathbf{x}^*, \quad (9)$$

sample $t \sim \mathcal{U}[0, 1]$, and form $\mathbf{x}_t = (1 - t)\mathbf{x}_0 + t\mathbf{x}_1$ with target velocity $\mathbf{v}_{\text{target}} = \mathbf{x}_1 - \mathbf{x}_0$. The Flow Matching loss directly supervises the network to output the correction velocity for both positions and yaws:

$$\mathcal{L}_{\text{flow}} = \mathbb{E}_{\mathbf{x}^*, \boldsymbol{\epsilon}, t} \left[ \| \mathbf{v}_\theta(\mathbf{x}_t, t, \mathcal{C}) - (\mathbf{x}_1 - \mathbf{x}_0) \|_2^2 \right]. \quad (10)$$

At inference, we deterministically correct the coarse pose by integrating the ODE $\frac{d\mathbf{x}(t)}{dt} = \mathbf{v}_\theta(\mathbf{x}(t), t, \mathcal{C})$ from $\mathbf{x}(0) = \mathbf{x}^c$ to obtain $\hat{\mathbf{x}} = \mathbf{x}(1)$, whose components yield the updated $\{(\hat{\mathbf{p}}_i, \hat{r}_i)\}_{i=1}^N$. Training with noisy $\mathbf{x}_0$ encourages learning a local correction field around the coarse estimate, while inference starts from the unperturbed coarse pose.

**Physical constraints via mesh-level SDF losses.** Relying solely on data-driven learning can still produce a few but fatal instances of interpenetration, directly causing physical simulation failures. To explicitly enforce simulation readiness, we augment the learning objective with differentiable mesh-level signed distance field (SDF) constraints. Compared to coarse bounding-box approximations, SDFs provide accurate shape boundaries and are particularly important for containment and stacking, where small pose errors can cause hidden intersections or unstable contacts.

For each mesh $m$, we precompute its SDF $D_m(\mathbf{x})$, where $D_m(\mathbf{x}) < 0$ indicates that $\mathbf{x}$ lies inside the mesh. For an object $i$, we sample a set of surface points $\mathcal{Q}_i$ on its mesh and define the signed penetration distance from $i$ to $m$ as

$$\text{dist}_{\text{sdf}}(i, m) = \min_{\mathbf{q} \in \mathcal{Q}_i} D_m(\mathbf{q}). \quad (11)$$

A negative value implies that some sampled points of object $i$ penetrate into $m$. We penalize interpenetration between any object pair $(i, j)$ via

$$\mathcal{L}_{\text{obj-obj}} = \sum_{i<j} [\max(0, -\text{dist}_{\text{sdf}}(i, j))]^2, \quad (12)$$

and prevent objects from penetrating the tabletop by modeling it as an SDF $\tau$:

$$\mathcal{L}_{\text{obj-table}} = \sum_i [\max(0, -\text{dist}_{\text{sdf}}(i, \tau))]^2. \quad (13)$$

In addition to forbidding penetration, we encourage stable resting contact to reduce floating artifacts and stabilize stacked configurations. For each object $i$, we sample points from its bottom region $\mathcal{B}_i$ and consider a set of candidate supports $\mathcal{S}_i$ (the tabletop and nearby objects after simple geometric filtering). For a candidate support $s \in \mathcal{S}_i$, we measure the distance from the bottom region to the support surface using the absolute SDF value:

$$\delta(i, s) = \min_{\mathbf{b} \in \mathcal{B}_i} |D_s(\mathbf{b})|. \quad (14)$$

We select the closest support $z_i^{\text{sup}} = \arg\min_{s \in \mathcal{S}_i} \delta(i, s)$ and define

$$\text{gap}(i, z_i^{\text{sup}}) = \min_{s \in \mathcal{S}_i} \delta(i, s). \quad (15)$$

The support-contact loss then enforces the bottom region to be within a small tolerance of the chosen support surface:

$$\mathcal{L}_{\text{sup}} = \sum_i [\max(0, \ \text{gap}(i, z_i^{\text{sup}}) - \epsilon)]^2. \quad (16)$$

Using $|D_s(\cdot)|$ penalizes both cases where the object floats above the support and where it lies inside concave supports but remains far from the supporting surface, thereby improving stability under complex containment and stacking. Finally, the overall training objective of the Physics Corrector is

$$\mathcal{L}_{\text{PC}} = \mathcal{L}_{\text{flow}} + \lambda_{\text{sdf}}(\mathcal{L}_{\text{obj-obj}} + \mathcal{L}_{\text{obj-table}}) + \lambda_{\text{sup}}\mathcal{L}_{\text{sup}}.$$

### 3.4. Dual-System Inference Pipeline

Algorithm 1 summarizes our progressive dual-system inference with a batched, pipelined schedule. For each scene, the Semantic Reasoner expands the layout in stages $(O^t \rightarrow O^B \rightarrow O^b)$, strengthening instruction grounding and reducing missing task-critical objects. After each semantic expansion, the Physics Corrector updates only translations and yaw rotations $(\mathbf{p}, r)$. Importantly, because SR and PC are decoupled modules, we can pipeline inference across a batch: when one scene is undergoing pose correction, other scenes can concurrently advance their semantic expansion, avoiding idle waiting between the two systems and improving throughput. The PC-corrected layout is fed back as context for the next SR stage, preventing geometric errors from compounding during later object placement. The procedure terminates after the final stage $O^b$ for each scene, yielding complete simulation-ready layouts. When the batch size reduces to $M=1$, Algorithm 1 degenerates to the standard serial alternation between SR and PC for a single scene.

## 4. Experiment

### 4.1. Experiment setup

**Dataset.** We build our training data on MesaTask-10K, which contains 10,000 tabletop scenes paired with task in-

*Table 1.* **Quantitative comparison** with baseline methods on task-driven tabletop layout generation. STABLE achieves the best generation performance on all evaluation metrics, consistently outperforming other baselines.

| Method | FID↓ | GPT Score | | | | | | OC | AwT(%) | AwS(%) |
|---|---|---|---|---|---|---|---|---|---|---|
| | | CwT | OSR | PPI | LCR | OV | Avg. | | | |
| GPT-4o | 85.5 | 6.0 | 8.0 | 8.4 | 6.2 | 6.9 | 7.1 | 25.4 | 25.2 | 18.5 |
| Holodeck-Table | 90.2 | 4.2 | 7.0 | 7.8 | 5.0 | 8.5 | 6.5 | 0 | 18.4 | 11.2 |
| I-Design-Table | 94.8 | 5.0 | 8.0 | 8.4 | 5.4 | 7.7 | 6.9 | 10.3 | 16.2 | 9.6 |
| MesaTask | 40.6 | 7.8 | 9.2 | 9.4 | 8.6 | 9.0 | 8.8 | 15.6 | 90.2 | 81.5 |
| TabletopGen | 54.8 | 8.6 | 9.3 | 9.5 | 8.9 | 8.7 | 9.0 | 23.5 | 91.7 | 83.6 |
| Steerable | 43.7 | 7.5 | 9.2 | 9.7 | 8.7 | 8.8 | 8.8 | 0 | 99.4 | 91.1 |
| Ours | 38.6 | 9.0 | 9.4 | 9.6 | 9.1 | 8.9 | 9.2 | 0 | 99.4 | 99.0 |

structions. For the Physics Corrector, we use all 10,000 scenes to train the geometry-aware pose correction model. For the Semantic Reasoner, we first follow the original MesaTask preprocessing to obtain 10,000 instruction–scene pairs. We then convert them into serialized multi-stage layout data according to 3.2, where each instruction is associated with three sub-layouts corresponding to $(O^t, O^t \cup O^B, O^t \cup O^B \cup O^b)$.

**Metrics.** Following prior work, we report FID to measure the visual realism of scenes and GPT-score to assess overall scene quality. To quantify task-conditioned correctness, we use Align with Task(AwT) and Align with Scene Graph(AwS), measuring how well the generated scene matches the input instruction. To evaluate simulation readiness, we report the physical feasibility metric Object Collision(OC), capturing inter-object penetrations.

**Baselines.** We compare STABLE with representative baselines spanning four categories of tabletop scene generation pipelines: (1) **Task-to-scene methods**, including MesaTask(Hao et al., 2026), I-Design-Table(Çelen et al., 2024), and Holodeck-Table(Yang et al., 2024); (2) **Post-processing baselines**, including MesaTask with refine and Steerable(Pfaff et al., 2025) PostProc. (3) **Proprietary models**, represented by GPT-4o(Achiam et al., 2023) under the same task-to-scene prompting and evaluation protocol; and (4) **Image-conditioned tabletop generation**, represented by TabletopGen(Wang et al., 2025);

### 4.2. Results

**Quantitative results.** As shown in Table 1, our method achieves the best overall performance across both semantic quality and physical feasibility. Compared to existing task-to-scene baselines, STABLE consistently improves visual realism and LLM-based semantic scores, while producing physically valid layouts with no collisions. In particular, we observe a clear trade-off in prior methods: LLM-centric approaches can sometimes generate plausible scenes but often suffer from collisions and weak task grounding, whereas

post-processing pipelines can eliminate collisions but may distort the original layout distribution and drift away from the task intent. For example, the Steerable post-processing baseline typically drives the collision metric to zero, yet it often resolves penetrations by aggressively relocating objects, disrupting relative spatial relations and substantially reducing task alignment. MesaTask exhibits stronger task alignment than most baselines, but still produces non-negligible physical violations. We also compare with an image-conditioned pipeline. Although TabletopGen uses images as input, complex tabletop relationships introduce visual ambiguities that make small objects hard to detect and can lead to error accumulation, resulting in weaker alignment with the task and scene graph than STABLE.

**Qualitative results.** Fig. 3 presents qualitative comparisons with representative baselines I-Design-Table and Holodeck-Table often generate sparse layouts and rarely capture vertical arrangements, which can lead to Misaligned Placement in stacking-oriented scenes. In containment, they also exhibit Physical Failure, most notably interpenetrations.

TabletopGen benefits from strong image priors, but the intermediate image representation is a bottleneck: small or thin items are easily missed, causing Missing Object, and occlusions or embedded stacking (e.g., multiple items inside a bowl) obscure depth and contacts, leading to Misaligned Placement and occasional Physical Failure after lifting to 3D. MesaTask can produce richer scenes, but still suffers from Missing Object. Moreover, MesaTask post-hoc refinement and Steerable reduces hysical Failure but may introduce Misaligned Placement by relocating objects and breaking intended relations (e.g., shifting an object off its specified support). In contrast, STABLE avoids these failure modes by alternating semantic scene construction with physics-aware pose correction, producing task-complete layouts with stable, simulation-ready geometry.

**Convergence robustness vs. post-hoc optimization.** We compare our Physics Corrector (PC) with the post-hoc optimization procedure used in MesaTask and Steerable under

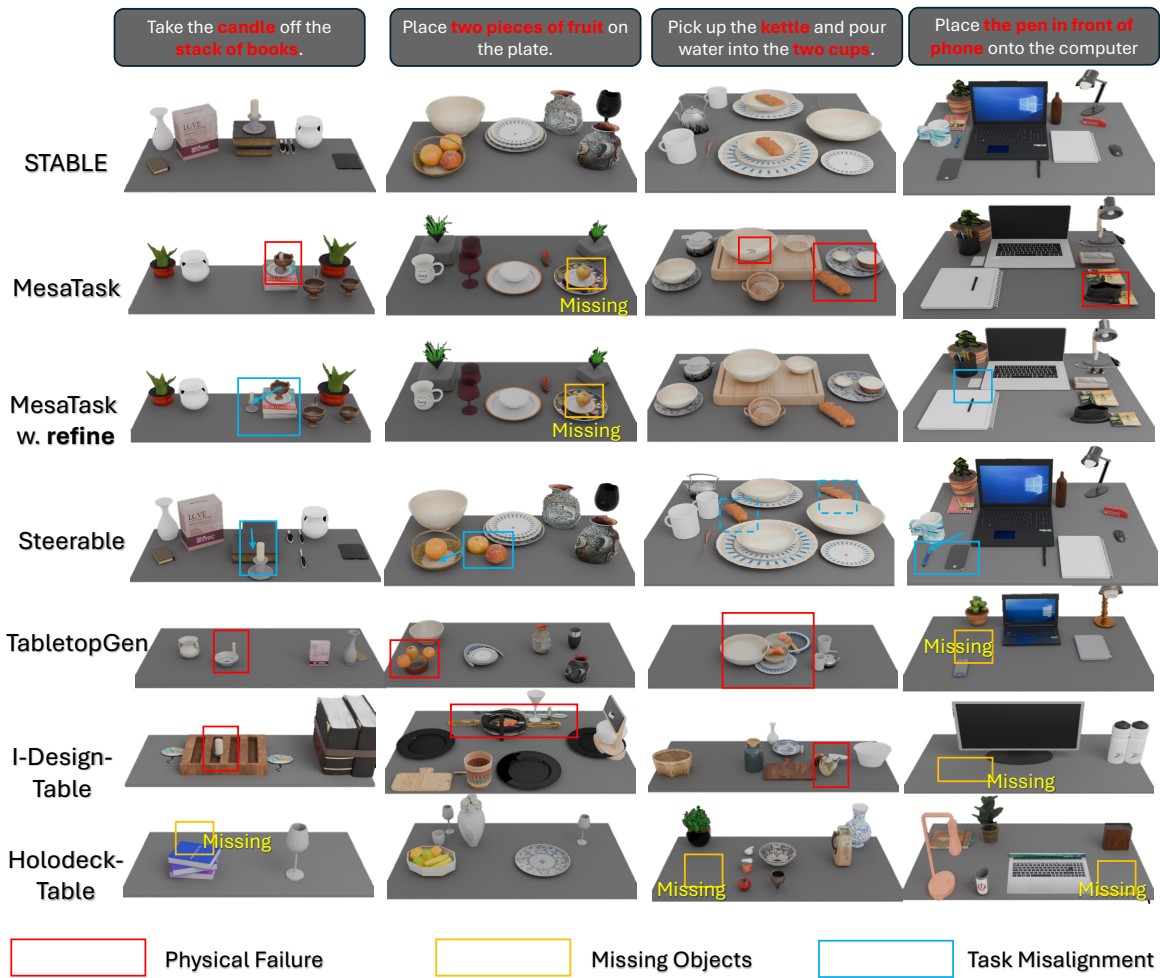

*Figure 3.* **Qualitative comparison** of task-conditioned tabletop scenes generated by STABLE and baselines. Red/yellow/blue boxes denote Physical Failure, Missing Objects, and Task Misalignment, respectively. STABLE yields task-aligned and physically plausible, simulation-ready layouts.

*Table 2.* **Collision-resolution robustness of the Physics Corrector.** We compare the MesaTask post-hoc optimization and Steerable with our Physics Corrector on initial layouts grouped by collision counts.

| Method | 0-10 | 10-20 | 20-30 | 30-40 |
|---|---|---|---|---|
| MesaTask's Optim. | ✓ | ✓ | ✗ | ✗ |
| Steerable | ✓ | ✓ | ✗ | ✗ |
| Physics Corrector | ✓ | ✓ | ✓ | ✓ |

different levels of initial collisions. We bucket test scenes by the number of object-object collisions in the input layout, and evaluate whether each method can recover a collision-free layout. For these optimizers, we allow a very large budget of 50,000 optimization iterations. As shown in Table 2, these optimizer succeeds when the initial collision level is low, but their success rate drops sharply as collisions become severe, often failing to reach a feasible collision-free configuration even under this strong budget. In contrast, our learned PC consistently produces collision-free layouts

across all collision regimes. These results highlight that learning-based pose correction is substantially more reliable than iterative post-hoc optimizations.

### 4.3. Ablation study

**What is the effect of physical constraints?** We study the contribution of each physical constraint in the Physics Corrector by ablating one loss term at a time while keeping all other components and training settings unchanged.

As shown in Table 3, removing any single constraint significantly degrades simulation readiness, indicating that the three losses are complementary. Removing the support-contact loss $\mathcal{L}_{\text{sup}}$ leads to a clear increase in floating artifacts, suggesting that explicit contact supervision is important for stabilizing resting and stacking behaviors. Removing the object–table collision loss $\mathcal{L}_{\text{obj-table}}$ causes a substantial rise in collisions; interestingly, the floating rate decreases in this setting, as objects tend to "resolve" support

*Table 3.* **Comprehensive Ablation Study.** We evaluate the physical constraints in the Physics Corrector and the progressive scene construction in the Semantic Reasoner.

| Variant | OC ↓ | Float ↓ | AwT ↑ | Distractor Rate ↑ |
|---|---|---|---|---|
| *(a) Ablation on Physical Constraints (Physics Corrector)* | | | | |
| w/o $\mathcal{L}_{\mathrm{sup}}$ | 4.7 | 9.8 | - | - |
| w/o $\mathcal{L}_{\mathrm{obj\text{-}table}}$ | 13.6 | 5.4 | - | - |
| w/o $\mathcal{L}_{\mathrm{obj\text{-}obj}}$ | 11.9 | 15.8 | - | - |
| Full (Ours) | 0 | 0 | - | - |
| *(b) Ablation on Progressive Scene Construction (Semantic Reasoner)* | | | | |
| One-shot SR | - | - | 89.9 | 78.6 |
| Progressive SR (Ours) | - | - | 99.4 | 86.1 |

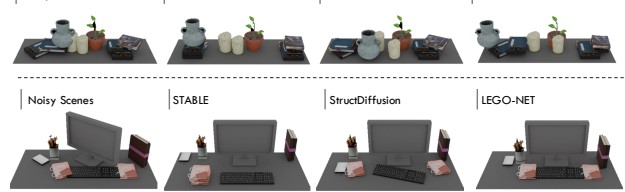

*Figure 4.* **Results of Editing.** Without any additional fine-tuning on edit-specific supervision, our STABLE Semantic Reasoner supports intuitive multi-step edits by locally modifying the structured layout while preserving overall scene semantics.

violations by sinking into the tabletop when penetration is no longer penalized. Finally, removing the object–object collision loss $\mathcal{L}_{\mathrm{obj\text{-}obj}}$ results in the most severe physical degradation overall, with both collision failures and floating artifacts increasing sharply due to widespread inter-object interpenetration and destabilized support relationships. Overall, these results highlight that collision avoidance and support consistency must be jointly enforced to obtain physically plausible layouts.

**What is the effect of progressive scene construction?** We study the impact of progressive layout construction in the Semantic Reasoner by comparing a one-shot variant that generates the full object set in a single pass with a progressive variant that expands the layout from task-critical objects to background distractors in multiple stages. As shown in Table 3, the progressive construction approach exhibits higher AwT, demonstrating a significant improvement in task alignment, indicating more reliable instruction grounding and fewer missing or misplaced task-critical objects. Meanwhile, it also increases the distractor rate, i.e., a larger fraction of generated objects are background clutter, suggesting that the staged formulation encourages richer scene completion rather than stopping at a minimal task-only layout. Overall, decomposing scene construction into stages helps the model better ground task semantics while producing more realistic, cluttered tabletop scenes.

### 4.4. Application

**Editing.** Beyond scene generation, our dual-system design also endows the Semantic Reasoner with strong editing ca-

*Table 4.* **Quantitative comparisons** on the task of Rearrangement

| Method | Distance Move ↓ | EMD to GT ↓ | OC ↓ |
|---|---|---|---|
| LEGO-NET | 0.28 | 0.43 | 0.32 |
| StructDiffusion | 0.21 | 0.23 | 0.25 |
| Ours | 0.14 | 0.08 | 0 |

*Figure 5.* **Qualitative comparison on Rearrangement.** Compared with StructDiffusion and LEGO-NET, STABLE produces more physically consistent layouts and better preserves functional relations under clutter.

pability, without additional fine-tuning on editing-specific supervision. This largely stems from its progressive layout construction: the Semantic Reasoner incrementally composes a scene by first reasoning about task-critical objects and then extending to contextual distractors. Such staged generation encourages the model to internalize object roles and inter-object relations in the layout; combined with the generalization ability of the base LLM, it enables flexible instruction-following edits at inference time. As shown in Fig. 4, the Semantic Reasoner can perform intuitive scene edits by modifying or re-completing parts of the structured layout while preserving overall task semantics.

**Rearrangement.** To evaluate the Physics Corrector on a downstream rearrangement task, we compare against LEGO-NET, an indoor scene rearrangement method, and StructDiffusion, a tabletop scene rearrangement method. As shown in Table 4 and Fig. 5, our method recovers more reasonable and physically consistent tabletop layouts than existing approaches. Details are provided in the appendix B.3.

## 5. Conclusion

In this paper, we present **STABLE**, a semantics–physics dual-system for simulation-ready tabletop scene generation. STABLE decouples semantic layout generation via an LLM-based Semantic Reasoner from physics-aware pose correction via a geometry-aware Physics Corrector, overcoming the 3D spatial reasoning limitations of LLM-only methods and the task misalignment issues of post-hoc optimization. Its progressive inference paradigm, alternating between the two systems, ensures scene generation expands from task-critical to background objects while maintaining the physical plausibility of scenes. Extensive experiments validate that STABLE outperforms baselines in both physical validity and task alignment, demonstrating robust capability in generating simulation-ready tabletop scenes.

## Acknowledgements

This work was supported by the Shanghai Municipal Special Program for Basic Research on General AI Foundation Models (Grant No. 2025SHZDZX025G02), in collaboration with Shanghai Artificial Intelligence Laboratory.

## Impact Statement

This paper presents work whose goal is to advance the field of Machine Learning. There are many potential societal consequences of our work, none which we feel must be specifically highlighted here.

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

# A. Training Details

**Semantic Reasoner Training.** We fine-tune Qwen3-8B with supervised fine-tuning (SFT) on our serialized instruction-to-layout data. We use a learning rate of $1 \times 10^{-5}$, a maximum sequence length of 5,000 tokens, and a micro-batch size of 4 per GPU. Training is performed for 1 epoch.

**Physics Corrector Training.** We train the Physics Corrector using Flow Matching (**?**) with linear interpolation paths to learn the vector field that transports samples from a standard Gaussian prior to the data distribution. The denoising network is a 1D U-Net with a hidden dimension of 512 and self-conditioning. Each object is represented as a 4D vector (3D position + z-rotation), conditioned on 64-dimensional point cloud features and learnable instance embeddings. During training, we use Adam optimizer with a learning rate of 2e-4 and batch size of 2,048 for 5,000 epochs. To encourage physically plausible layouts, we add mesh-level SDF constraints to the training objective, including object–object collision loss ($\lambda_{\text{sdf}} = 0.02$), object–table collision loss ($\lambda_{\text{sdf}} = 0.02$), and support-contact loss ($\lambda_{\text{sup}} = 0.01$).

# B. Details of Experiment

### B.1. Metrics

**FID** Following MesaTask(Hao et al., 2026), measures visual realism between rendered and real scenes.

**GPT-Score.** We follow the GPT-based multi-criteria evaluation protocol introduced in MesaTask to assess semantic alignment and perceptual quality of generated tabletop scenes. For each scene, we render both a front view and a perspective view, and prompt a pretrained LLM to jointly consider the rendered images together with the corresponding task description (including environment/task context). The model rates the scene along five criteria: (1) Consistency with Task, (2) Object Size Reasonableness, (3) Placement Plausibility & Intersections, (4) Layout Coherence & Realism, and (5) Object Visibility. Each criterion is scored on a 1–10 scale and accompanied by a brief rationale. We use MesaTask's structured prompt design to encourage consistent scoring across scenes, and report the per-criterion scores as well as their average. We use the same prompting template and rendering protocol as MesaTask whenever applicable.

**Object Collision (OC).** We quantify physical validity using the object collision rate. For a generated scene with $N$ objects, we consider all unordered object pairs $\{(i,j)|i < j\}$ as potential collision pairs. We detect collisions using mesh-level SDF queries: if the signed distance between two objects is negative, i.e., $\text{SD}(O_i, O_j) < 0$, we treat $(i,j)$ as a colliding pair. The OC score is defined as the number of colliding pairs normalized by the total number of potential pairs:

$$\text{OC} = \frac{\sum_{i<j} \mathbb{I}\left[\text{SD}(O_i, O_j) < 0\right]}{\binom{N}{2}}.$$

**Align with Task (AwT).** AwT measures whether the generated scene contains the required task-critical objects specified by the instruction. Let $\mathcal{O}_{\text{gt}}^{\star}$ be the set of task objects required by the ground-truth annotation and $\mathcal{O}_{\text{pred}}^{\star}$ be the set of task objects generated by the method. We compute

$$\text{AwT} = \frac{\left|\mathcal{O}_{\text{pred}}^{\star} \cap \mathcal{O}_{\text{gt}}^{\star}\right|}{\left|\mathcal{O}_{\text{gt}}^{\star}\right|}.$$

**Align with Scene Graph (AwS).** AwS evaluates the distribution of task-relevant objects based on the annotated scene graph, checking whether it conforms to the task instructions. Let $\mathcal{S}_{\text{gt}}$ represent the set of scene graphs between task objects in the ground truth scene graph, and let $\mathcal{S}_{\text{pred}}$ represent the set of scene graphs between the generated task objects (matched by object category/identity). We define

$$\text{AwS} = \frac{|\mathcal{S}_{\text{pred}} \cap \mathcal{S}_{\text{gt}}|}{|\mathcal{S}_{\text{gt}}|}.$$

Both AwT and AwS are reported as ratios in $[0, 1]$, where higher values indicate better semantic completeness.

**Floating rate (Float).** We additionally report Float to quantify whether objects are stably supported. For each object $i$, we first identify its supporting surface $z_i^{\text{sup}}$ (either the tabletop or another object) using the same support detection procedure as in the support-contact loss $\mathcal{L}_{\text{sup}}$. We then compute the mesh-level bottom-to-support separation $g(i)$ by querying the SDF of the selected support at the bottom region of object $i$ (following the distance computation used in $\mathcal{L}_{\text{sup}}$). An object is

considered floating if its separation exceeds a small threshold $\delta_{\text{float}}$:

$$\mathbb{I}_{\text{float}}(i) = \mathbb{I}[g(i) > \delta_{\text{float}}], \qquad \delta_{\text{float}} = 0.05 \text{ cm}. \tag{17}$$

The Float metric is defined as the percentage of floating objects in a scene, averaged over the test set:

$$\text{Float} = \frac{1}{|\mathcal{D}|} \sum_{J \in \mathcal{D}} \frac{1}{N_J} \sum_{i=1}^{N_J} \mathbb{I}_{\text{float}}(i), \tag{18}$$

where $N_J$ is the number of objects in scene $J$ and $\mathcal{D}$ denotes the evaluation set.

### B.2. Baselines

We compare against four categories of baselines and provide implementation details here. Unless otherwise specified, we follow MesaTask (Hao et al., 2026) for prompt templates, rendering settings, and evaluation protocols whenever applicable.

#### B.2.1. TASK-TO-SCENE BASELINES.

**MesaTask.** We evaluate the original MesaTask pipeline (Hao et al., 2026) as a representative task-to-scene method. MesaTask generates a structured tabletop layout from the task instruction and retrieves 3D assets accordingly. We use the official preprocessing and evaluation protocol provided by MesaTask.

**Holodeck-Table.** We adopt the tabletop adaptation of HOLODECK (Yang et al., 2024) provided in MesaTask. Concretely, the pipeline first queries a closed-source LLM to propose the object set and their coarse spatial relations, then retrieves corresponding 3D assets, and finally applies an optimization-based placement search to obtain a plausible tabletop layout. Following MesaTask, we tailor the prompting and remove room-specific modules that are irrelevant to tabletop scenes (e.g., wall/window-related components). In addition, we use the same tabletop-oriented constraints in the optimization stage to account for differences between room-scale and tabletop-scale layouts (e.g., desktop objects are typically not anchored to walls and should avoid unrealistic edge-biased placements).

**I-Design-Table.** We also follow MesaTask's adaptation of I-Design (Çelen et al., 2024). The baseline uses a multi-agent LLM setup to translate language input into a feasible scene graph that specifies relative spatial relations among objects, and then employs a backtracking-based solver to place objects accordingly. As noted in MesaTask, I-Design's optimization stage is lightweight and transfers to tabletop settings with minimal algorithmic changes; therefore, we mainly adapt the prompt to the tabletop domain while keeping the original placement procedure intact.

#### B.2.2. IMAGE-CONDITIONED TABLETOP GENERATION.

**TabletopGen.** We evaluate TabletopGen (Wang et al., 2025) as a representative image-conditioned pipeline. It synthesizes a 2D image representation of the target tabletop layout and then lifts it to a 3D scene by predicting object identities and poses from the image. For a fair comparison, we follow the same rendering and asset library as in our evaluation protocol when converting the predicted layout into a 3D scene.

#### B.2.3. PROPRIETARY MODELS.

**GPT-4o.** To contextualize performance against strong general-purpose systems under the same task-to-scene input setting, we evaluate GPT-4o using the same task prompts and the same evaluation protocol. We prompt the model to output a structured layout in our JSON format and then run the same asset retrieval and rendering pipeline for metric computation.

#### B.2.4. POST-PROCESSING BASELINES.

In addition to end-to-end generation baselines, we include post-processing methods that "sanitize" an initial layout.

**MesaTask+refine.** We apply MesaTask's optimization-based post-processing module (Hao et al., 2026) to the layouts generated by MesaTask, following the same solver configuration and stopping criteria as in the original implementation.

**Steerable.** We also compare against a steerable post-processing baseline (Pfaff et al., 2025), where we first use our Semantic Reasoner to generate a coarse (potentially colliding) layout from the task instruction and then feed this layout into the steerable post-processing module to produce a physically feasible layout. This baseline isolates the effect of post hoc correction when the initial layout is provided by a strong semantic generator.

---

**Algorithm 1** Dual-System Inference Loop

---

**Require:** A batch of task instructions $\{I^{(m)}\}_{m=1}^M$; tabletop specs $\{T^{(m)}\}_{m=1}^M$; Semantic Reasoner SR; Physics Corrector PC; stage
    schedule $\mathcal{K} = [t, B, b]$
**Ensure:** Simulation-ready layouts $\{J^{(m)}\}_{m=1}^M$
 1: For each scene $m$: initialize $\mathcal{O}^{(m)} \leftarrow \emptyset$, $J^{(m)} \leftarrow \{T^{(m)}, \emptyset\}$, stage index $\kappa^{(m)} \leftarrow 1$
 2: Initialize two queues: $Q_{SR} \leftarrow \{1, \ldots, M\}$, $Q_{PC} \leftarrow \emptyset$
 3: **while** exists $m$ with $\kappa^{(m)} \leq |\mathcal{K}|$ **do**
 4:    **if** $Q_{SR}$ is not empty **and** SR resources available **then**
 5:       Pop an index $m$ from $Q_{SR}$
 6:       $k \leftarrow \mathcal{K}[\kappa^{(m)}]$                                     ▷ Current stage for scene $m$
 7:       $\Delta\mathcal{O}^{(m)} \leftarrow \text{SR}(I^{(m)}, T^{(m)}, J^{(m)};\, k)$
 8:       $\mathcal{O}^{(m)} \leftarrow \mathcal{O}^{(m)} \cup \Delta\mathcal{O}^{(m)}$
 9:       $J^{(m)} \leftarrow \{T^{(m)}, \mathcal{O}^{(m)}\}$                         ▷ Update semantics; keep attributes fixed
10:       Push $m$ into $Q_{PC}$                             ▷ Pose correction for the updated layout
11:    **end if**
12:    **if** $Q_{PC}$ is not empty **and** PC resources available **then**
13:       Pop an index $m$ from $Q_{PC}$
14:       $\{(\mathbf{p}_i^{(m)}, r_i^{(m)})\}_{O_i \in \mathcal{O}^{(m)}} \leftarrow \text{PC}(J^{(m)})$
15:       Update poses in $J^{(m)}$ using $\{(\mathbf{p}_i^{(m)}, r_i^{(m)})\}$
16:       $\kappa^{(m)} \leftarrow \kappa^{(m)} + 1$                                   ▷ Advance to next stage
17:       **if** $\kappa^{(m)} \leq |\mathcal{K}|$ **then**
18:          Push $m$ into $Q_{SR}$                         ▷ Feed PC-corrected layout back to SR
19:       **end if**
20:    **end if**
21: **end while**
22: **return** $\{J^{(m)}\}_{m=1}^M$

---

## B.3. Details of Rearrangement

We using the same 500 test samples described above. We construct rearrangement inputs by perturbing the translation $\mathbf{p}$ and yaw rotation $r$ of all objects in each scene with Gaussian noise of standard deviation 0.1. In our rearrangement experiments, we report two complementary metrics that capture both the magnitude of changes and the accuracy of recovery. Distance Moved measures the average displacement introduced by a rearrangement method: we first establish one-to-one correspondences between same-category objects in the perturbed input and the rearranged output using Earth Mover's Distance (EMD), then compute the Euclidean distance of each matched pair, average over objects within each scene, and finally average over all test scenes. EMD to GT evaluates how closely the rearranged scene matches the ground-truth (GT) canonical layout: treating the predicted and GT object sets as two distributions, we compute EMD over a joint feature space consisting of 3D translation $t_i$ and rotation $r_i$, and report the mean transport cost across the test set.

## C. Flow Matching Details for Pose Correction

We adopt conditional Flow Matching to learn a continuous-time pose correction dynamics anchored to the coarse layout predicted by the Semantic Reasoner. Let $\mathbf{x} = [\mathbf{p}_1, \ldots, \mathbf{p}_N, r_1, \ldots, r_N]$ denote the pose vector (translations and yaw angles) for $N$ objects, and let the conditioning be $\mathcal{C} = (\mathbf{x}^c, \mathbf{G})$, where $\mathbf{x}^c$ is the coarse pose and $\mathbf{G}$ are geometry embeddings of retrieved assets.

**Endpoint construction.** Unlike unconditional generative models that start from pure noise, pose correction should remain close to the coarse layout. Therefore, we construct the noise endpoint by perturbing the coarse pose: $\mathbf{x}_0 = \mathbf{x}^c + \sigma\epsilon$, $\epsilon \sim \mathcal{N}(\mathbf{0}, \mathbf{I})$, and set the data endpoint to the ground-truth pose $\mathbf{x}_1 = \mathbf{x}^*$. This design makes the learned flow explicitly model a refinement trajectory from a noisy coarse estimate toward a physically valid pose.

**Interpolation path and target velocity.** We use the standard linear path between endpoints: $\mathbf{x}_t = (1 - t)\mathbf{x}_0 + t\mathbf{x}_1$, $t \sim \mathcal{U}[0, 1]$. Under this path, the target velocity field is constant: $\mathbf{v}_{\text{target}} = \frac{d\mathbf{x}_t}{dt} = \mathbf{x}_1 - \mathbf{x}_0$. We parameterize the conditional velocity with a neural network $\mathbf{v}_\theta(\mathbf{x}_t, t, \mathcal{C})$ and minimize

$$\mathcal{L}_{\text{flow}} = \mathbb{E}_{\mathbf{x}^*, \epsilon, t}\left[\left\|\mathbf{v}_\theta(\mathbf{x}_t, t, \mathcal{C}) - (\mathbf{x}_1 - \mathbf{x}_0)\right\|_2^2\right]. \tag{19}$$

*Table 5.* Manipulability detection results on generated scenes.

| Object | AP@IoU=0.5 | AP@IoU=0.75 |
|--------|-----------|-------------|
| kettle | 0.87 | 0.65 |
| fork | 0.91 | 0.71 |

**TV Table**

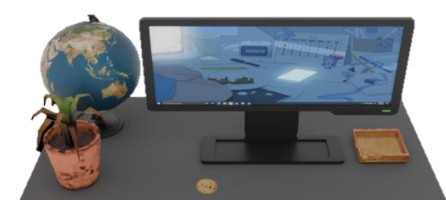

**On the TV table**, move the **potted plant** to the **front of** the **TV**, leaving a small gap

**Side Table**

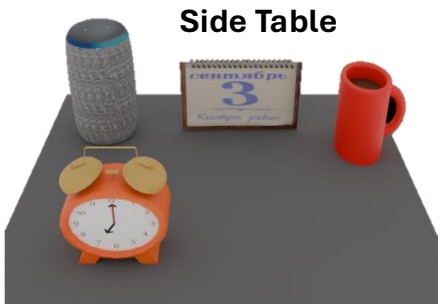

**On the side table**, move the **alarm clock** to the **right of** the **desk calendar**

*Figure 6.* **Generalization Results.** STABLE generalizes to unseen tabletop types, producing well-structured and collision-free layouts.

**Deterministic inference via ODE integration.** At test time, we perform deterministic correction by solving the ODE $\frac{d\mathbf{x}(t)}{dt} = \mathbf{v}_\theta(\mathbf{x}(t), t, \mathcal{C})$ from $\mathbf{x}(0) = \mathbf{x}^c$ to obtain $\hat{\mathbf{x}} = \mathbf{x}(1)$. In practice, we use a standard numerical solver with a fixed small number of steps, yielding stable and efficient refinement.

**Practical notes.** (1) The noise scale $\sigma$ controls correction locality: larger $\sigma$ encourages stronger edits but may destabilize fine contacts. (2) Yaw angles are treated consistently during training/inference (e.g., wrapping to a canonical range or using an equivalent continuous representation) to avoid discontinuities at $2\pi$. (3) Conditioning $\mathcal{C}$ is injected through the network (e.g., concatenation/FiLM/cross-attention), enabling geometry-aware refinement while preserving object identities and sizes.

## D. Additional Generalization and Controllability Experiments

We provide additional qualitative experiments to further evaluate the generalization and controllability of STABLE beyond the standard MesaTask-10K evaluation setting. These experiments focus on three out-of-distribution cases: unseen tabletop types, unseen object assets, and ambiguous or conflicting task instructions.

### D.1. Generalization to Unseen Tabletop Types

To evaluate whether STABLE generalizes beyond the tabletop categories observed during training, we test it on unseen tabletop categories that are not included in MesaTask-10K, including TV tables, side tables, nightstands, and TV stands. For these unseen tabletop categories, we use GPT-4o to generate corresponding task-oriented instructions and evaluate STABLE under the same generation setting as in the main experiments.

As shown in Fig. 6 and Fig. 7, STABLE can generate coherent, task-aligned, and simulation-ready layouts on these out-of-distribution tabletop categories. The generated scenes preserve plausible object selections and spatial arrangements for different tabletop functions, while the Physics Corrector continues to remove physical violations such as collisions and floating objects. These results suggest that STABLE is not limited to the tabletop categories in MesaTask-10K and can transfer to new support surfaces with different sizes, shapes, and functional contexts.

### D.2. Generalization to Unseen Object Assets

We further evaluate whether the Physics Corrector can handle object geometries that are not present in the original asset library. To this end, we introduce 100 new high-quality object assets generated by Hunyuan3D. During test-time asset retrieval, we first search within this new asset set and only fall back to the original asset library when no suitable match is

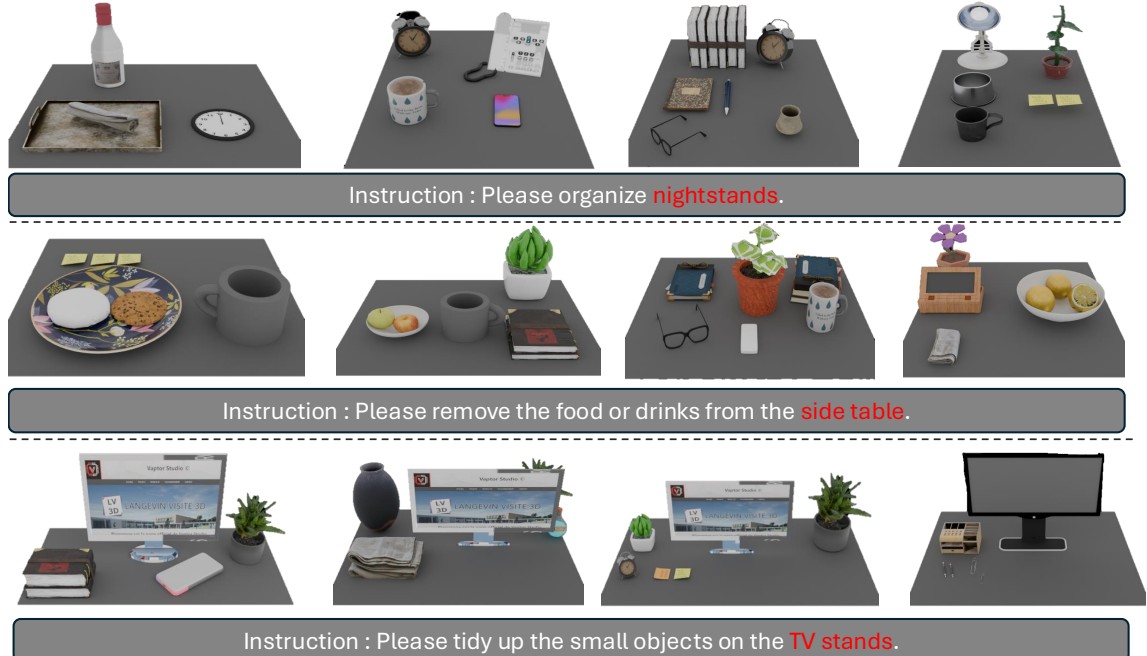

*Figure 7.* **Generalization to unseen tabletop types.** STABLE generalizes to tabletop types that are not included in MesaTask-10K, including nightstands, TV stands, and side tables. For each unseen tabletop type, task instructions are generated by GPT-4o. STABLE produces coherent, task-aligned, and physically plausible layouts on these out-of-distribution support surfaces.

found.

As shown in Fig. 8, STABLE remains effective when using these unseen assets. Although the new assets introduce different mesh geometries, shapes, and aspect ratios, the Physics Corrector can still produce physically plausible placements by leveraging geometry-aware conditioning. This indicates that the corrector does not simply memorize the training asset library, but instead learns transferable pose correction behavior conditioned on object geometry.

### D.3. Controllability under Ambiguous or Conflicting Instructions

Our standard setting takes a single task-oriented instruction as input for each scene. These instructions are designed to describe plausible robotic manipulation tasks, so strongly conflicting spatial constraints are relatively uncommon in MesaTask-10K. Nevertheless, we additionally test STABLE on deliberately ambiguous or conflicting instructions to examine its controllability under out-of-distribution language inputs.

As shown in Fig. 9, STABLE tends to follow such instructions in a largely literal and controllable manner, even when the requested layouts are unusual, such as placing a plate above a cup or putting headphones on top of a keyboard. We do not claim that STABLE contains a dedicated conflict-resolution mechanism for arbitrary inconsistent instructions. Instead, these results suggest that the Semantic Reasoner can expose controllable semantic intent in the structured layout, while the Physics Corrector attempts to maintain physical feasibility without changing object identities or task-relevant relations.

## E. Additional ManipVQA Experiments

Our method can generate simulation-ready desktop scenes that can be directly used for physical simulations. To further evaluate the physical manipulability of the generated scenes, we introduced ManipVQA for robot manipulability detection experiments. Specifically, we selected two common graspable objects—kettles and forks—and sampled 50 complex scenes containing the target objects from the test set to construct two detection tasks: "detecting the graspable area of the kettle" and "detecting the graspable area of the fork." For annotation, we labeled the graspable areas of the kettle handle and the fork handle as ground-truth bounding boxes. The results in the table 5 show that ManipVQA's detection performance on our generated scenes is close to its performance on the original benchmark, indicating that existing manipulability detection models can be stably transferred to scenes generated by STABLE. This experiment verifies from a downstream perspective that the generated scenes not only satisfy geometric and physical feasibility constraints but also possess good

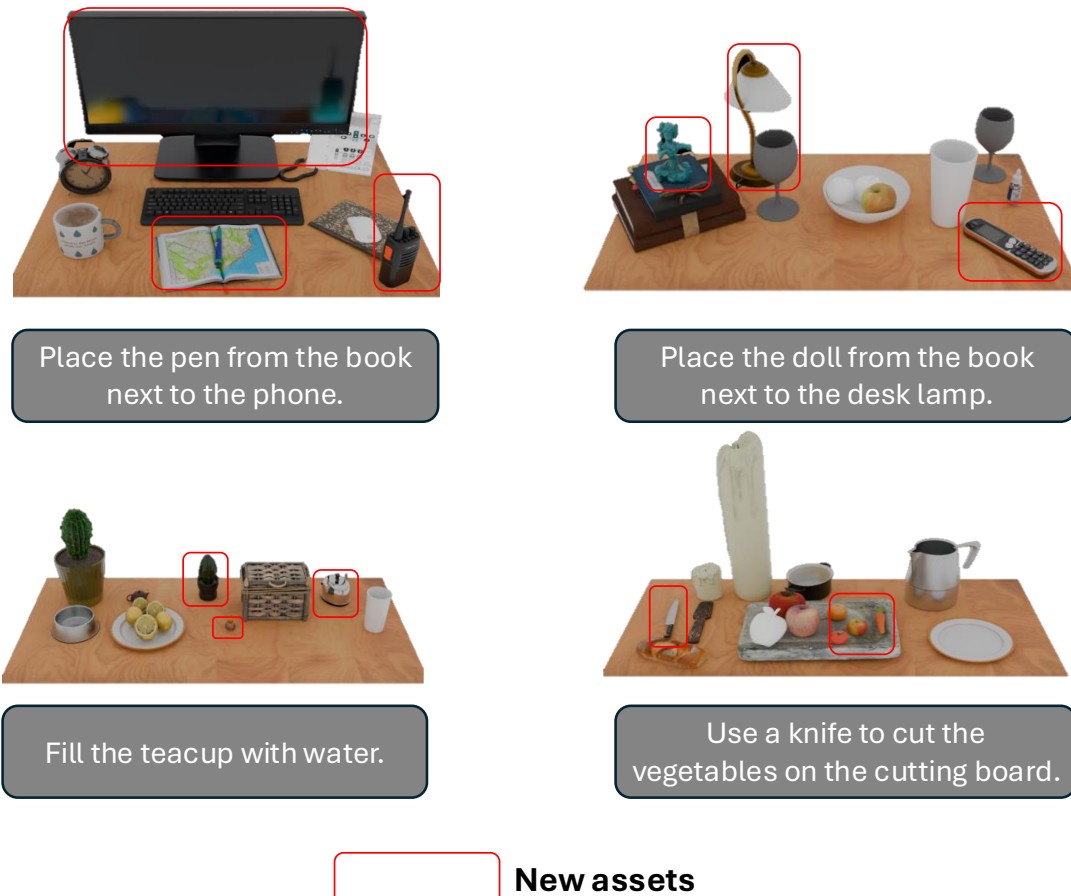

*Figure 8.* **Generalization to unseen object assets.** We introduce 100 new high-quality assets generated by Hunyuan3D and prioritize this new asset set during test-time retrieval. STABLE remains effective under these unseen geometries, suggesting that the Physics Corrector learns transferable geometry-aware pose correction rather than memorizing the original asset library.

robot manipulation usability, and can serve as a supplementary indicator for evaluating the physical realism of the scenes.

## F. More Results

We provide additional qualitative visualizations to complement the quantitative results in the main paper. Fig 10 illustrate its ability to produce diverse, cluttered tabletop scenes while maintaining simulation readiness under challenging configurations such as stacking and container-based placement. Fig 11 across representative task-to-scene pipelines, including Steerable, MesaTask, MesaTask w/ refinement, and STABLE. These comparisons highlight common failure modes of prior methods— e.g., physically invalid placements, and semantic drift introduced by post-processing—and demonstrate how STABLE better preserves task grounding while achieving physically consistent layouts.

## G. Limitations and Future Work

While STABLE achieves strong performance on simulation-ready tabletop layout generation, it has several limitations that suggest promising future directions. First, our current layout representation models object rotation as a single scalar (yaw) around the vertical axis. This assumption is reasonable for many tabletop assets but becomes restrictive for objects with non-upright resting poses, articulated parts, or tasks that require full 6D orientation and richer state variables (e.g., tilt, roll, joint states, open/closed states, and deformable configurations). Extending the Physics Corrector to predict full 6DoF poses and additional object states, together with state-aware physical constraints, would improve expressiveness and enable more complex manipulation scenarios.

Second, our progressive dual-system inference currently follows a fixed three-stage schedule ($O^t \rightarrow O^B \rightarrow O^b$). In

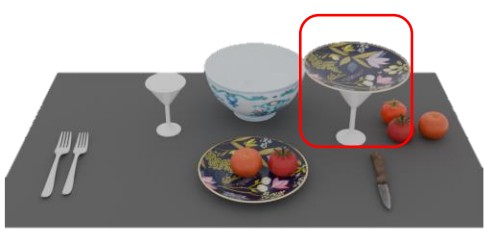

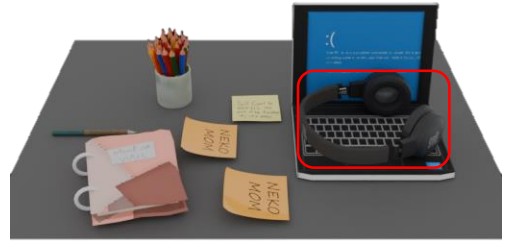

*Figure 9.* **Controllability under ambiguous or conflicting instructions.** We test STABLE with deliberately unusual or conflicting spatial constraints. Although STABLE is not designed as a dedicated conflict-resolution system, it follows the given instructions in a largely literal and controllable way while maintaining physically plausible layouts when possible.

principle, the same mechanism can be generalized to an unbounded number of rounds, allowing the Semantic Reasoner to continuously expand or revise a scene and the Physics Corrector to maintain physical feasibility throughout. Exploring adaptive stopping criteria, dynamic stage scheduling, and streaming-style generation could enable scalable scene expansion and more flexible incremental editing, especially for long-horizon scene construction.

Finally, STABLE is designed and evaluated on tabletop scenes. A natural next step is to extend the framework to larger indoor environments that contain multiple tabletop-like surfaces and more diverse support structures, such as shelves, cabinets, racks, and warehouse storage systems. These settings introduce additional challenges, including multi-surface support reasoning, long-range spatial constraints, and larger-scale asset diversity. We believe the semantics–physics decomposition in STABLE provides a strong foundation for scaling toward such complex indoor operational scenes.

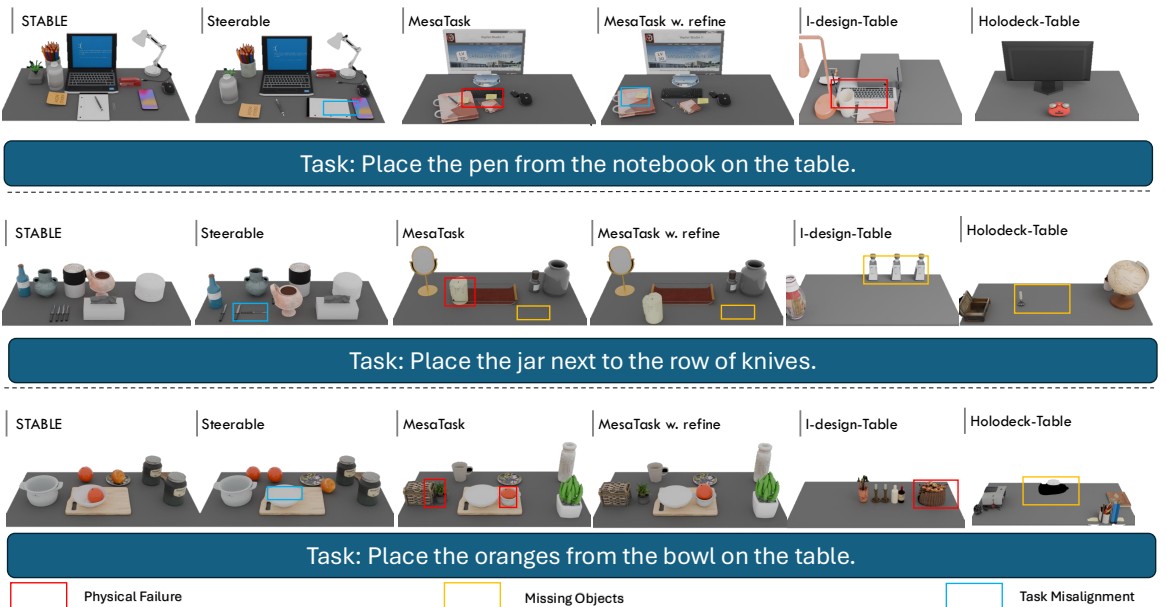

*Figure 10.* More qualitative comparisons of task to scene generation results across Steerable, MesaTask, MesaTask with refine and STABLE.

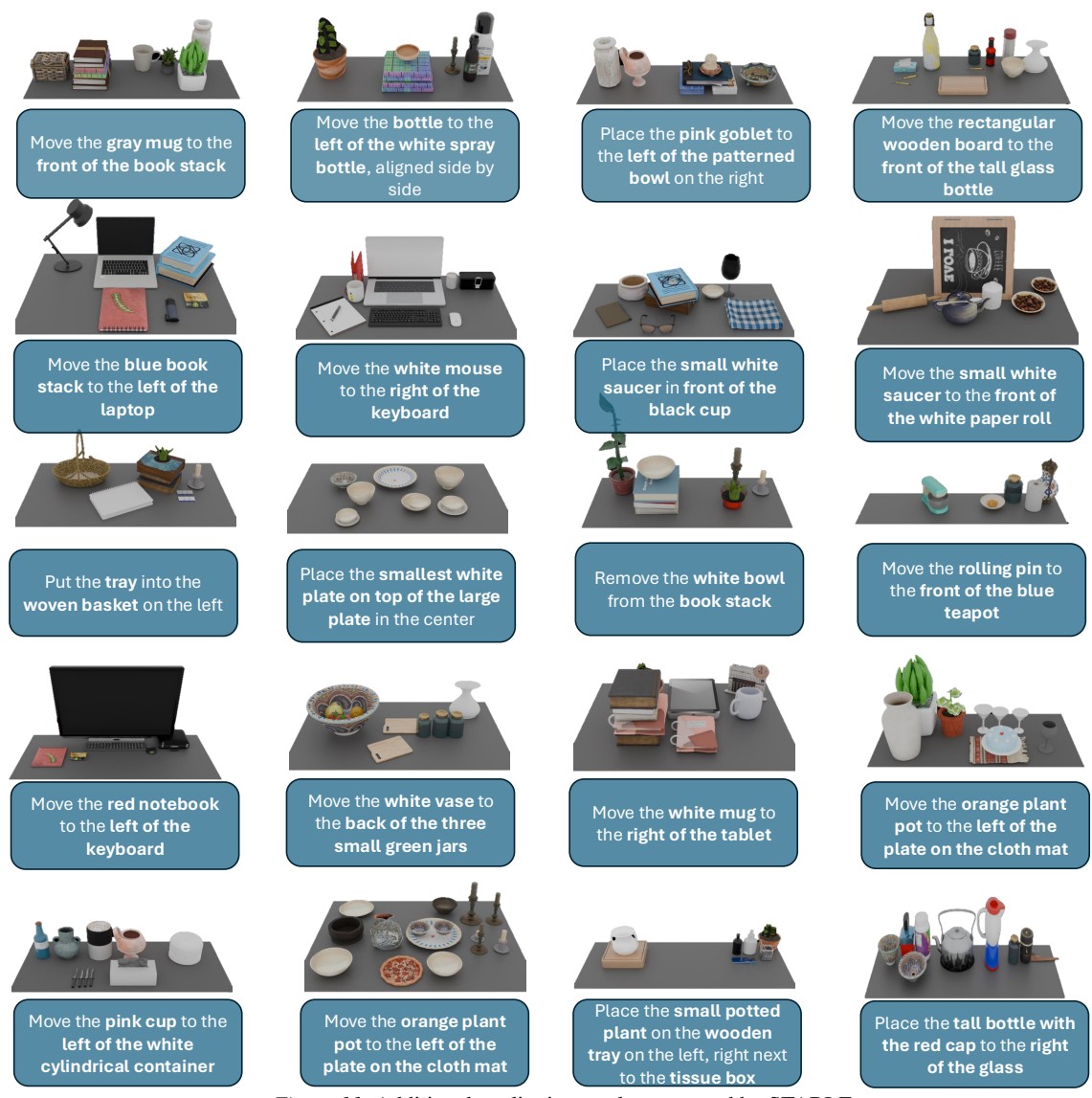

*Figure 11.* Additional qualitative results generated by STABLE.

