# OpenReview forum: "STABLE: Simulation-Ready Tabletop Layout Generation via a Semantics–Physics Dual System"
_ICML.cc/2026/Conference — ICML 2026 regular_

### Official Review · Reviewer_hb5o · 2026-03-06

**Soundness:** 2
**Presentation:** 3
**Significance:** 3
**Originality:** 2
**Overall Recommendation:** 4
**Confidence:** 3

**Summary:**

This paper introduces STABLE, a dual-system framework designed for generating simulation-ready tabletop scenes from task instructions. The system consists of two primary components: a Semantic Reasoner, which utilizes an LLM to interpret task instructions and generate coarse, task-aligned scene layouts, and a Physics Corrector, a flow-based denoising model that refines the poses of objects to ensure physical plausibility. The paper highlights STABLE's ability to generate realistic, collision-free, and task-specific scenes by alternating between these two components in a progressive inference process. The experiments show that STABLE outperforms existing methods in terms of both task alignment and physical feasibility.

**Compliance With Llm Reviewing Policy:**

Affirmed.

**Final Justification:**

The rebuttal has addressed my concerns, and I increase the score to 4.

**Key Questions For Authors:**

1. How does the performance of the model scale with increasingly complex and cluttered scenes? Are there specific configurations or environments where the system might fail to preserve physical plausibility or task alignment?

2. In terms of task alignment, how does the model handle instructions with ambiguous or conflicting requirements (e.g., conflicting spatial constraints for objects)? How robust is the system to such complexities in real-world use cases?

3. What are the model’s limitations when generating scenes for highly constrained tasks, like precise object placement in dense environments? How does the system handle these edge cases, and is there potential for further optimization?

4. Can the Semantic Reasoner be fine-tuned for new, unseen tasks or domains, and if so, how does the model maintain semantic consistency while expanding to new environments?

5. What is the impact of the physics-aware flow-based approach compared to more traditional physics engines or optimization techniques? How does the model's approach outperform other methods in terms of simulation readiness?

**Limitations:**

The authors state that “There are many potential societal consequences of our work, none which we feel must be specifically highlighted here”, yet this aspect ought to be elaborated on in the paper.

**Strengths And Weaknesses:**

**Strengths:**
The framework is technically feasible, integrating semantic and physics correction modules with a progressive generation process and mesh-level SDF loss functions to address issues with LLM-based methods and ensure the physical rationality of spatial relationships in generated scenes. The paper is well-organized and clearly expressed, boasting a solid theoretical basis, adequate references to prior research, detailed methodological explanations and sufficient experimental support, while its figures and diagrams also assist in understanding complex methodologies and scene generation processes. Moreover, the study explores task-oriented scene generation by combining semantic understanding and physical realism to fill a relevant research gap in this field, with its focus on simulation-readiness providing research references for embodied AI and robotic manipulation applications, and the proposed STABLE method offering a new approach to generating task-aligned, simulation-ready scene layouts, whose designed dual-system architecture and progressive generation paradigm further provide novel design insights for the scene generation domain.

**Weakness:**
1. While the Semantic Reasoner relies on LLMs to generate task-aligned layouts, LLMs are limited in 3D spatial reasoning. They struggle with continuous and dynamic geometric transformations, potentially causing errors in object placement, especially in complex spatial configurations. Although the Physics Corrector mitigates some of these issues, intricate stacking or dense interactions may still lead to edge-case failures not fully addressed in the paper.

2. The Physics Corrector is crucial for refining the Semantic Reasoner's outputs. However, in scenes with severe collisions or complex interactions, the Physics Corrector may struggle to find feasible solutions. Furthermore, it doesn’t address issues in cluttered or overlapping scenes, which may require additional optimization. The paper lacks a detailed discussion on how the system handles such complex configurations.

3. While the progressive inference process helps manage scene complexity, it doesn’t fully address the scalability issue as scene size and complexity increase. The system might struggle to maintain task alignment and physical feasibility with larger scenes or those requiring high-precision placement. The performance limits in such cases are not clearly explored in the paper.

4. The model relies heavily on the MesaTask-10K dataset, and it’s unclear how well STABLE generalizes to unseen tasks or domains. The system may struggle with new object categories or unfamiliar task configurations, potentially requiring further fine-tuning to handle novel scenarios effectively.

5. The Semantic Reasoner might face challenges when tasked with ambiguous or conflicting instructions (e.g., "place the plate to the right of the cup" vs. "place the plate above the cup"). The model may struggle to disambiguate such conflicts, affecting task alignment and scene quality. The paper doesn’t provide insight into how the system handles these complexities in real-world scenarios.

6. The two-stage refinement process (Semantic Reasoner + Physics Corrector) increases the computational cost. The Physics Corrector introduces significant overhead, and while the progressive generation reduces error accumulation, it remains unclear how the system performs under real-time or large-scale simulation requirements, particularly in resource-constrained environments.

---

> ### Author Rebuttal · Authors · 2026-03-31
>
> We sincerely thank you for the encouraging feedback and thoughtful summary.  We appreciate your recognition of dual-system architecture and progressive generation paradigm. Below, we address your comments and respond to the specific concerns you raised.
>
> **On challenging edge cases in highly cluttered and collision-heavy scenes**
>
> We agree that LLMs have inherent limitations in continuous 3D spatial reasoning, especially in complex scenes, which is exactly why our framework does not rely on the Semantic Reasoner alone. After task-specific fine-tuning, the Semantic Reasoner provides a task-aligned coarse layout, while the Physics Corrector refines it into a simulation-ready one by resolving collisions and enforcing support relations. Empirically, this design is effective for the vast majority of evaluation scenes and substantially improves physical validity over the Semantic Reasoner alone. We also agree that extremely dense interactions or intricate stacking may still present edge cases, and we are happy to clarify this limitation more explicitly in the revision. At the same time, our experimental setting suggests that, after fine-tuning, the main challenge is not uncontrolled geometric failure from the Semantic Reasoner, but rather the residual physical inconsistencies that can be effectively handled by the Physics Corrector in most cases.
>
> **On ambiguous or conflicting instructions**
>
> Our model takes a single task-oriented instruction as input for each scene, rather than multiple independent instructions that must be jointly reconciled. In our setting, such instructions are meant to describe plausible robotic manipulation tasks, so strongly conflicting spatial constraints are relatively uncommon. Still, we agree that these cases are worth testing. We therefore conducted additional qualitative experiments with deliberately conflicting instructions. As shown in the supplementary figure 3(https://anonymous.4open.science/api/repo/PDF-B12E/file/rebuttal1.pdf?v=b496899d), the model can still follow such constraints in a largely literal and controllable way, even when they lead to unusual layouts, such as placing a plate above a cup or headphones on top of a keyboard.. We do not claim a dedicated conflict-resolution mechanism for such out-of-distribution inputs, but the results suggest reasonable controllability beyond the main benchmark setting.
>
> **On adaptation to new tasks or domains**
>
> This concern is closely related to the generalization issue discussed above with Reviewer q68z. As noted there, we have already examined transfer to unseen tabletop types and unseen assets. This consistency mainly comes from the extrapolation ability of the underlying LLM, after task-specific fine-tuning, the SR learns to align task instructions with structured scene layouts in a way that generalizes beyond the training categories.
>
> **The physics-aware flow-based corrector versus traditional optimization**
>
> We note that this comparison is already partially included in the paper: Steerable is reported in the main results, and MesaTask + optimization is shown in the qualitative comparisons. These baselines were included to contrast our physics-aware flow-based corrector with traditional optimization-based refinement. The key difference is that optimization-based methods rely on iterative test-time correction, which can be computationally heavy, sensitive to initialization, and prone to distorting scene semantics under severe collisions. In contrast, our Physics Corrector is a learned refinement model that directly maps coarse semantic layouts to physically improved ones, leading to better simulation readiness in practice, as reflected by the stronger physical validity and more stable layouts in our experiments. We agree that this comparison can be highlighted more clearly in the revision.
>
> **On computational cost under real-time or large-scale settings**
>
> Our method is not designed for real-time scene generation during robot simulation. Instead, the goal is to generate large numbers of simulation-ready scenes offline for large-scale simulation. In this setting, scene generation and simulation are decoupled, so real-time responsiveness is not the target use case. As noted in the teaser, generating one scene currently takes about 60 seconds, including three SR stages and three PC stages. We will clarify this scope more explicitly in the revision.
>
> **Societal impact**
>
> We agree that the societal impact discussion should be more explicit. While our work has positive applications in embodied AI, capable scene-generation systems may also reduce some manual labor in scene assembly and synthetic data construction, which are still partly performed by artists or content workers in existing pipelines. If extended beyond research settings, similar methods could also affect parts of interior or furniture design workflows by automating preliminary layout generation. We will revise the paper to discuss these potential impacts more clearly.

---

> > ### Author Rebuttal · Reviewer_hb5o · 2026-04-03
> >
> > Thanks for the response; my concerns have been mostly addressed. I increase my score to 4.

---

> > > ### Author Response · Authors · 2026-04-07
> > >
> > > Thank you very much for your thoughtful response and for updating your score. We sincerely appreciate your positive assessment and are glad that our rebuttal was able to address your concerns. Thank you again for your constructive and encouraging review.

---

### Official Review · Reviewer_q68z · 2026-03-10

**Soundness:** 3
**Presentation:** 3
**Significance:** 3
**Originality:** 2
**Overall Recommendation:** 4
**Confidence:** 4

**Summary:**

This paper proposes STABLE, a semantics-physics dual-system framework designed to address the physical implausibility commonly found in tabletop scenes generated by Large Language Models. The method consists of an LLM-based semantic reasoner that progressively generates coarse layouts, and a Flow-Matching-based Physics Corrector that refines object poses using mesh-level signed distance function constraints. Experiments on the MesaTask dataset demonstrate that STABLE can generate simulation-ready tabletop scenes that strictly align with task instructions while maintaining physical validity.

**Compliance With Llm Reviewing Policy:**

Affirmed.

**Final Justification:**

The rebuttal has addressed my concerns, and I therefore maintain my positive rating.

**Key Questions For Authors:**

Please refer to the weaknesses.

**Limitations:**

Yes

**Strengths And Weaknesses:**

**Strengths**:
1. The authors propose a principled manner to separate semantic reasoning from geometric correction. The decomposition aligns well with known limitations of LLMs in geometric reasoning.
2. The physics-aware correction module is technically well designed. The integration of flow matching for pose updates is an interesting design choice that enables continuous correction in pose space.
3. The experimental section is reasonably comprehensive and demonstrate the effectiveness of the proposed method.


**Weaknesses**:
1. The overall methodological novelty appears somewhat overstated. The idea of combining an LLM-based semantic planner with a downstream geometric or physics-based correction module has appeared in multiple forms in embodied AI and scene synthesis pipelines. The paper frames the approach as a “first-of-its-kind dual-system framework,” but this claim seems overstated given prior work on physics-aware optimization or modular planning execution pipelines.

2. The Semantic Reasoner itself is relatively standard. It is essentially a fine-tuned LLM that outputs structured layouts, and the main modification is a staged serialization of object groups. This component does not introduce fundamentally new modeling techniques and mainly relies on dataset restructuring.

3. The reliance on the MesaTask-10K dataset raises concerns about generalization. The entire training pipeline appears tightly coupled to this dataset and its annotation structure. There is no evaluation on a different dataset or asset library to demonstrate robustness across domains.

4. The progressive generation strategy is reasonable but somewhat heuristic.

5. Physical validity metrics are relatively limited. Although collision rate and floating rate are measured, the evaluation does not include actual physics simulation stability.

6. The computational cost of the dual-system inference pipeline is not clearly analyzed. Alternating between LLM inference, asset retrieval, and pose correction may introduce latency.

---

> ### Author Rebuttal · Authors · 2026-03-31
>
> We thank you for the clear summary and positive feedback.  We appreciate your recognition of our motivation, modular design. Below, we address your concerns and provide clarifications on the specific points you raised.
>
> **The scope of the claimed methodological novelty**
>
> We agree that simply combining an LLM-based semantic module with a downstream geometric or physics-based correction module is not entirely new. Our novelty claim is more specific: to the best of our knowledge at the time of submission, STABLE is the first dual-system framework for scene generation in which two functionally distinct subsystems interact progressively during inference, rather than operating as a one-pass two-stage pipeline. This distinction matters. In a two-stage pipeline, a full layout is generated first and corrected only afterward. In STABLE, the scene is expanded stage by stage, and the Physics Corrector is applied after each stage before the next semantic expansion. Thus, our “dual-system” claim refers to this interactive alternating design, not merely the use of two modules. We are happy to revise the wording to avoid overstating the claim, and we would appreciate it if the reviewer could point us to the specific prior works they have in mind so that we can cite them appropriately.
>
> **The novelty of the Semantic Reasoner**
>
> We agree that the Semantic Reasoner itself is not intended to be a fundamentally new modeling technique. It is a fine-tuned LLM for structured layout generation, and we do not claim it as a standalone contribution. In our paper, its role is to serve as the semantic component of the overall dual-system framework, leveraging the generalization and instruction-following ability of LLMs to align task instructions with scene layouts. Existing works such as LLM-based layout generation pipelines have already shown the promise of this direction, and our goal is not to reinvent that component in isolation, but to build on it and address its key limitations, which we discuss in the paper, especially its difficulty in ensuring physical validity and simulation readiness. In this sense, the Semantic Reasoner should be understood as one part of the full system design, rather than as the main source of novelty on its own.
>
> **Generalization on unknown desktop types and new object assets**
>
> First, we test generalization to unseen tabletop types that are not included in MesaTask-10K. In the supplementary material, we already showed several initial examples, and we further expand this analysis with additional results on nightstands, TV stands, and side tables, which are outside the tabletop categories in MesaTask-10K. For these unseen tabletop types, we use GPT-4o to generate corresponding task instructions and evaluate our pipeline under the same generation setting. The qualitative results figure 1, provided in the supplementary link(https://anonymous.4open.science/api/repo/PDF-B12E/file/rebuttal1.pdf?v=b496899d), show that our method can still generate coherent and simulation-ready layouts on these unseen surfaces.
> Second, we test generalization to unseen assets by introducing 100 new high-quality objects generated by Hunyuan3D. During retrieval at test time, we first search within this new asset set, and only fall back to the full original library if no suitable match is found. The supplementary link results figure 2 show that the Physics Corrector remains effective under these unseen assets(https://anonymous.4open.science/api/repo/PDF-B12E/file/rebuttal1.pdf?v=b496899d). Together, these results suggest that our framework generalizes beyond the original MesaTask-10K tabletop categories and asset library.
>
>
> **Physics simulation stability beyond collision and floating metrics**
> We agree that collision rate and floating rate alone do not fully capture physics simulation stability. Therefore, we conducted an experiment to verify the stability of the generated scene in the simulation, as we discussed with Reviewer mvi7 above.
>
>
> **On the computational cost of the dual-system inference pipeline**
>
> We agree that alternating between LLM inference, asset retrieval, and pose correction can introduce additional latency compared with a single-pass pipeline. To mitigate this, our implementation is not purely sequential in practice. Instead, it is designed for asynchronous execution, so that during batch generation, different modules can work concurrently rather than remaining idle. In particular, asset retrieval and Physics Corrector computation for one batch of partial layouts can proceed while the Semantic Reasoner processes the next batch, improving hardware utilization and reducing idle waiting. Thus, although the dual-system design introduces more stages, its latency is explicitly mitigated through asynchronous scheduling and parallel utilization. We will clarify this inference-time tradeoff more explicitly in the revision.

---

> > ### Author Rebuttal · Reviewer_q68z · 2026-04-03
> >
> > Thank the authors for the detailed rebuttal. I appreciate your candid acknowledgement of the method's limitations, and your additional efforts, have successfully resolved several of my initial concerns. I highly encourage you to incorporate these discussions and the new generalization results into the final revision of the manuscript. Given the constructive response, I tend to keep my positive score.

---

> > > ### Author Response · Authors · 2026-04-07
> > >
> > > Thank you very much for your thoughtful and encouraging response. We sincerely appreciate your acknowledgment that our rebuttal has addressed several of your initial concerns, as well as your helpful suggestion to incorporate the additional discussion and generalization results into the final revision. We will certainly do so. We are also grateful for your positive view of the paper. If you feel that the added clarifications and evidence further strengthen the work, we would be very grateful if this could be taken into account in your final assessment. Thank you again for your constructive and valuable review.

---

### Official Review · Reviewer_mvi7 · 2026-03-12

**Soundness:** 3
**Presentation:** 3
**Significance:** 3
**Originality:** 3
**Overall Recommendation:** 4
**Confidence:** 4

**Summary:**

This paper introduces STABLE, a semantics–physics dual-system designed for generating simulation-ready tabletop scenes. STABLE comprises two primary modules: a semantic reasoner, represented by a fine-tuned LLM that outputs coarse scene layouts according to high-level task instructions, and a physics corrector, which is a geometry-aware, flow-based denoising model that refines object poses to ensure physical plausibility while preserving semantic intent. The system employs a progressive generation strategy, alternately invoking the semantic and physics modules to expand scenes iteratively, starting with essential objects and gradually adding contextual elements. Experimental results include both quantitative and qualitative analyses, aiming to demonstrate the effectiveness of STABLE in generating physically plausible and semantically aligned scenes.

**Compliance With Llm Reviewing Policy:**

Affirmed.

**Final Justification:**

Thanks for the effort on the rebuttal. The response has fully addressed my concerns. Hence, I am inclined to recommend acceptance of this work.

**Key Questions For Authors:**

As mentioned in the weaknesses.

**Limitations:**

As mentioned in the weaknesses.

**Strengths And Weaknesses:**

## Strengths
1. The proposed semantics–physics dual-system is reasonable and contributes to improved robustness in tabletop scene generation tasks.
2. The introduction of a geometry-aware denoising target within the physics corrector is novel and interesting.
3. Both quantitative and qualitative experimental analysis are satisfactory and demonstrate the practical potential of the approach.

## Weaknesses
1. The claim that object interpenetration or floating artifacts are solely due to LLM hallucinations requires further explanation and examples. More detailed reasoning about how these errors arise, and how the proposed method addresses them, would improve readability and credibility. Furthermore, LLM hallucinations are likely not the only cause of such artifacts.
2. While the use of GPT-based scores as a primary evaluation metric is acceptable, given the system’s consideration of physical attributes, it would be appropriate to introduce additional scientifically grounded graphics metrics as part of the evaluation standard.
3. The paper should report optimization time. It is unclear whether the dual-system architecture significantly increases overall simulation time, which could be unacceptable under large-scale settings. Optimization efficiency is critical for practical deployment.
4. Figure 2 needs improvement. The visual quality is low, the font is too small, and there are inconsistencies in font casing and formatting.
5. In Table 3, the proposed method reports zero occurrences of object interpenetration and floating artifacts (“oc” and “float”). The reliability of this result should be clarified, as such flawless performance is unusual unless the test set is extremely small. Further explanation is needed.

---

> ### Author Rebuttal · Authors · 2026-03-31
>
> We sincerely thank you for the positive feedback. We appreciate your recognition of the effectiveness of our semantic-physical dual system and the novelty of our physical corrector. Below we address your questions.
>
> **Causes of interpenetration and floating artifacts**
>
> We thank the reviewer for this important clarification. We agree that object interpenetration or floating artifacts should not be attributed solely to LLM hallucinations. Our intended point is that these artifacts arise because an LLM is fundamentally a probabilistic sequence model that learns to fit the distribution of scene descriptions and layouts in data, but does not explicitly model continuous 3D geometry, contact constraints, or physical feasibility. As a result, an LLM may produce layouts that are semantically plausible at a symbolic level, yet still violate the geometric precision required for simulation readiness.
>
> **On additional physically grounded evaluation metrics**
>
> Beyond GPT-based scores and basic physical validity metrics, we argue that additional evidence-based evaluations of simulation behavior would make this paper more thorough. To provide a stronger evaluation, we conducted an additional experiment in Isaac Sim: each generated scene is imported into the simulator, assigned default physical properties, simulated for 10 seconds, and evaluated by the average object displacement before and after simulation. As shown below,
> | Metrics | Ours | Steerable | TabletopGen | MesaTask | I-Design-Table | Holodeck-Table | GPT-4o | Gemini | Claude |
> |---|---:|---:|---:|---:|---:|---:|---:|---:|---:|
> | Move Distances | 1.2 | 0.9 | 21.7 | 18.2 | 5.8 | 0.6 | 25.9 | 17.3 | 19.6 |
>
> methods with severe collisions exhibit much larger motion, while our method remains substantially more stable. We also observe that some methods with zero OC still show non-zero displacement; based on our inspection, this is mainly due to asset geometry and default physics parameters rather than the layout generation method itself. For example, top-heavy objects may tip over under uniform density, and assets with uneven bottoms may wobble slightly. We further note that our method is not the absolute lowest among near-zero-OC methods, likely because our scenes are generally richer and contain more objects. However, the overall gap remains small, and the results still support the strong simulation stability of our method.
>
> **On optimization time and large-scale efficiency**
>
> This concern is closely related to the efficiency question raised by reviewer gX78, which we address above.
>
> **On the presentation quality of Figure 2**
>
> We thank the reviewer for this suggestion. We agree that Figure 2 can be improved in terms of visual quality and readability. In the revision, we will increase the figure resolution, enlarge the font size, and unify the font casing and overall formatting for better consistency.
>
> **The experimental setting and metric computation for OC / Float**
>
> We thank the reviewer for this important question. We would like to clarify that the zero OC / Float values in Table 3 are computed using the automatic geometric metrics defined in the appendix: OC is based on mesh-level SDF collision queries, and Float is based on bottom-to-support separation with a fixed threshold, averaged over the evaluation set. We also agree that the test-set size was not stated clearly enough in the paper. This was an oversight on our side. In the current version, the appendix specifies the test-set size only in the rearrangement setting, while the number of evaluation scenes for the main generation results (including Table 3) was not explicitly restated. We will make this clearer in the revision. For the main evaluation, we use the same number of test scenes as MesaTask, namely 500 scenes under the same protocol.

---

> > ### Author Rebuttal · Reviewer_mvi7 · 2026-04-01
> >
> > Thanks for the effort on the rebuttal. The response has fully addressed my concerns. Hence, I am inclined to recommend acceptance of this work.

---

> > > ### Author Response · Authors · 2026-04-07
> > >
> > > Thank you very much for your positive assessment and for indicating that your concerns have been fully addressed. We sincerely appreciate your thoughtful and constructive feedback, which has helped us strengthen both the clarity and completeness of the paper. We are pleased that the additional clarifications and experiments in our rebuttal were useful. If you feel these additions further strengthen the work, we would be grateful if that could be reflected in your final assessment. Thank you again for your valuable and encouraging review.

---

### Official Review · Reviewer_gX78 · 2026-03-13

**Soundness:** 3
**Presentation:** 3
**Significance:** 2
**Originality:** 2
**Overall Recommendation:** 4
**Confidence:** 3

**Summary:**

STABLE presents how to generate physics constraints aware layout generation. As much as the idea is interesting, at the core there is an LLM based reasoning module and corrector which is a denoising model. It uses mesh-level signed distance function (SDF) collision losses and support-contact losses to enforce physical plausibility without disrupting the semantic alignment established by the Reasoner. The authors note specificity of this problem and rotation across different axes issue.

**Compliance With Llm Reviewing Policy:**

Affirmed.

**Final Justification:**

My questions were addressed to some extent

**Key Questions For Authors:**

- Objects are partitioned into three groups ($O^t, O^B, O^b$) based on a "predefined threshold" for bounding box intersections. The specific value of this threshold or how sensitive the progressive generation is to this choice is not specified.
- While the formulation states that asset identifiers ($a_i$) are retrieved from a library using size ($s_i$) and description ($d_i$), the specific retrieval mechanism or how it handles potential mismatches between LLM-predicted sizes and actual available mesh geometries is not detailed.
- More comparisons to models such as claude 4.6/ gemini 3.1 can be helpful
- A discussion on efficiency will add some interest

**Limitations:**

See Questions

**Strengths And Weaknesses:**

While the method looks technically sound, this is a very specific use case thereby limiting the significance of the problem. The amount of innovation seems too much for a specific problem and hence a cost analysis w.r.t other methods is highly recommended. The work seems to be original but presentation could be improved by following simple flow of Problem->Solution->Impact which is missing in this case. Although the authors discuss the problems such as rotation, it remains a large gap to understand why this big pipeline is needed

---

> ### Author Rebuttal · Authors · 2026-03-30
>
> We sincerely thank the reviewer for recognizing the interesting problem setting, the technical soundness of our method, and the originality of our approach.
>
> **Our writing flow and the reclaim  of our pipeline**
>
> We believe this concern mainly stems from a misunderstanding of the target problem setting rather than the method itself. Our goal is not generic tabletop layout generation, but task-conditioned simulation-ready scene generation, which is important for embodied AI because tabletop scenes are widely used for manipulation. This problem is challenging because even small pose errors can make a scene physically invalid. For this reason, our pipeline is not unnecessarily large, but motivated by the problem structure itself. The Semantic Reasoner handles instruction grounding and coarse scene composition, while the Physics Corrector enforces physical validity; their progressive interaction prevents errors from accumulating during scene construction. The SR is also important for rotation-related issues: due to biased probabilistic priors in training data, LLM-based methods often produce object orientations that are misaligned with real-world usage, as also noted in prior works such as OptiScene[1] and SceneWeaver[2]. Our SR improves these coarse semantic and rotational priors, providing a stronger initialization for downstream physics-aware correction. Thus, the design reflects the dual requirement of semantic correctness and simulation readiness, rather than unnecessary methodological complexity.
>
> [1] Yang et al. Optiscene: Llm-driven indoor scene layout generation via scaled human-aligned data synthesis and multi-stage preference optimization. (NeurIPS 2025)
> [2] Yang et al. Sceneweaver: All-in-one 3d scene synthesis with an extensible and self-reflective agent. (NeurIPS 2025)
>
> **Sensitivity analysis of the predefined threshold used for partitioning object groups**
>
> The predefined threshold is used in the data reconstruction step to partition objects into $O^t, O^B, O^b$, and is not a particularly sensitive model design choice. In this paper, we use a threshold of 2 cm. We agree that clarifying its value and sensitivity would make the paper more complete. To make this clearer, we additionally tested 1 cm, 4 cm, and 8 cm by reconstructing the dataset under each setting. The results show only minor differences. This suggests that our method is not highly sensitive to the threshold, and that the paper’s main contribution of simulation-ready scene generation holds reliably across sensible design choices.
> | | FID↓ | GPT Score | OC | AwT(%) | AwS(%) |
> |-|-|-|-|-|-|
> | 0.01 | 39.2 | 9.0 | 0 | 99.7 | 99.3 |
> | 0.02 | 38.6 | 9.2 | 0 | 99.4 | 99.0 |
> | 0.04 | 38.1 | 8.9 | 0 | 99.1 | 98.5 |
> | 0.08 | 39.5 | 9.1 | 0 | 98.7 | 97.2 |
>
> **On the asset retrieval mechanism and geometry mismatch handling**
>
> Our asset retrieval strategy is based on Holodeck and MesaTask. We first use Sentence-BERT to match the generated description $d_i$ with asset descriptions in the library, and keep the top-10 most semantically similar candidates. We then compare their bounding-box size ratios with the predicted size $s_i$, select the closest one, and resize the retrieved asset to $s_i$. This two-step design helps avoid semantically correct but geometrically mismatched assets, which could otherwise produce unrealistic results after resizing.
>
> **Additional comparisons with other closed-source LLMs**
>
> We already include GPT-4o as a strong closed-source baseline, and we agree that additional frontier models further strengthen the evaluation. Following the reviewer’s suggestion, we additionally test Gemini and Claude. As shown below,
> | | FID↓ | GPT Score | OC | AwT(%) | AwS(%) |
> |-|-|-|-|-|-|
> | Gemini | 61.3 | 7.9 | 18.0 | 34.1 | 28.3 |
> | Claude | 78.6 | 7.7 | 20.3 | 28.5 | 21.0 |
>
> They follow a similar trend to GPT-4o: while semantically plausible at a coarse level, they still struggle with the geometric precision and physical consistency required for simulation-ready scene generation.
>
> **Discussion on efficiency**
>
> We agree that efficiency should be discussed more explicitly. In our current implementation, generating one scene takes about 60 seconds, as noted in the teaser, including three SR stages and three PC stages. The progressive design keeps this cost manageable, since each PC step only refines a partial sub-layout rather than a fully cluttered scene. Our experiments also suggest a practical efficiency advantage: the paper shows that optimization-based post-hoc refinement can remain ineffective even with a very large optimization budget under severe collisions, indicating poor efficiency in complex scenes. In contrast, our Physics Corrector is a learned refinement module with predictable inference cost. During batched generation, our asynchronous implementation further improves throughput by overlapping retrieval and PC execution with subsequent SR computation. We will make this efficiency perspective clearer in the revision.

---

> > ### Author Rebuttal · Reviewer_gX78 · 2026-04-02
> >
> > i think my answers were kindof resolved. Alttough I would have like to see some numbers and experiments on efficiency but that is fine for now. I hope this will resolved in case the paper is accepted

---

> > > ### Author Response · Authors · 2026-04-07
> > >
> > > Thank you very much for your positive assessment and for confirming that your concerns have been fully resolved. We truly appreciate your thoughtful, constructive, and encouraging feedback. Your questions and suggestions have helped us improve the paper’s clarity, presentation, and empirical support. We are especially glad that the additional clarifications and experiments in our rebuttal proved helpful. Thank you again for your valuable review.

---

### Decision · Program_Chairs · 2026-04-30

**Decision:**

Accept (regular)

**Comment:**

This paper was reviewed by four experts in the field, all of whom provided Weak Accept recommendations. Based on the overall reviewer feedback, the decision is to recommend acceptance of the paper. The reviewers nevertheless identified several important issues that should be addressed in the camera-ready version, particularly the need for more detailed experimental comparisons and ablation studies, as raised by Reviewers gX78 and mvi7, and improvements to the clarity of the presentation and problem statement, as noted by all four reviewers. The authors are encouraged to revise the paper accordingly to the best of their ability.